# WHITE-BOX PROMPT TRANSFORMERS: VARIATIONALLY GROUNDED PROMPT–ATTENTION COUPLING FOR UNIFIED IMAGE RESTORATION

## ABSTRACT

**Can soft prompts in vision Transformers be made explainable?** Prompt-based models have achieved remarkable success in image restoration, yet they remain largely opaque: the underlying Transformer operations and the mechanism by which prompts modulate attention are poorly understood. This work revisits *guided image restoration*, where an auxiliary modality $A$ assists in restoring a target modality $B$. We interpret $A$ as a prompt and formulate a tailored structure-tensor total variation (*STV*) model, whose gradient suggests a white-box correspondence to prompt–attention interactions. This provides a principled bridge between prompts and attention. In scenarios where $A$ is unavailable, we abstract its role into learnable soft prompts, enabling end-to-end training within standard Transformer pipelines. By unrolling the gradient flow of the *STV* variational problem, we derive the *White-Box Prompt Transformer (WBPT)*, a cascaded architecture that embeds interpretability directly into attention operations. Extensive experiments on multiple benchmarks demonstrate that WBPT achieves state-of-the-art restoration performance while offering interpretable, controllable, and robust prompt–attention dynamics.

## 1 INTRODUCTION

Prompt-based Transformers have recently reshaped unified image restoration, enabling a single model to tackle diverse degradations through learnable soft prompts (Potlapalli et al., 2023). These prompts condition the restoration process by modulating attention mechanisms and consistently deliver strong empirical results (e.g., Jia et al., 2022; Kong et al., 2025). However, despite their success, prompt-based designs remain fundamentally opaque: the inner workings of the Transformer and the interaction between prompts and attention lack interpretability, limiting both theoretical understanding and practical controllability (Chefer et al., 2021; Jain & Wallace, 2019). This opacity impedes reliable deployment in trust-sensitive applications (Rudin, 2019).

This motivates our central question:

*Can prompt-driven attention be explained from first principles, providing a theoretically grounded interpretation of the black box?*

We draw inspiration from *guided image restoration*, where an auxiliary modality $A$ (e.g., $T_1$-weighted MRI) provides structural guidance for restoring a target modality $B$ (He et al., 2012; Li et al., 2016; Ehrhardt & Betcke, 2016a). In this setting, $A$ acts as a prior, naturally analogous to a prompt guiding the restoration of $B$ (Jia et al., 2022; Potlapalli et al., 2023). Since explicit auxiliary data are often unavailable (Havaei et al., 2016), we abstract the role of $A$ into learnable soft prompts—trainable tokens that emulate auxiliary guidance through end-to-end optimization. This perspective reinterprets prompts not as heuristic inputs but as principled surrogates for classical guidance (Li & Liang, 2021; Zhou et al., 2022).

Building on this analogy, we introduce a variational perspective on prompt-based restoration. Specifically, we cast guided restoration as a *structure-tensor total variation (STV)* problem (Chambolle & Pock, 2011; Lefkimmiatis et al., 2015). Through gradient analysis, we show that the optimization dynamics naturally align with a white-box attention mechanism, suggesting a formal link

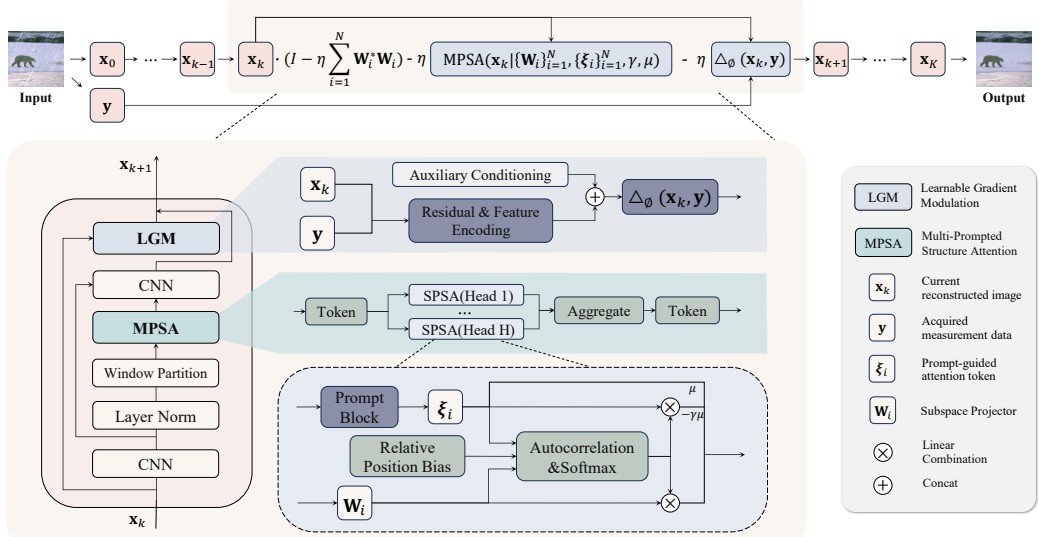

Figure 1: Overview of the White-box Prompt Transformer (WBPT). Image restoration is achieved by unrolling $K$ gradient-flow steps. At iteration $k$ (Eq. 8), the update combines a Multi-Prompted Structure Attention (MPSA) block with a learnable gradient-modulation (LGM) data-consistency term. MPSA consists of Single-Prompted Structure Attention (SPSA) heads: tokenized features interact with prompt tokens $\boldsymbol{\xi}_i$ and learnable projectors $\mathbf{W}_i$, are aggregated, and mapped back to image domain. Across stages, prompts interact with features at multiple levels, enriching structural context while preserving fidelity to the measurements.

between guidance priors (or their prompt surrogates) and Transformer attention (Yu et al., 2023; Wang et al., 2018; Meng et al., 2024). While the derivation involves approximations, it provides a principled foundation for understanding and designing interpretable prompt-driven restoration.

Crucially, this formulation not only offers interpretability but also suggests a concrete architectural design. By unrolling the gradient flow of the *STV* variational problem, we obtain the *White-Box Prompt Transformer (WBPT)* (Chen et al., 2015; Monga et al., 2021). Each Transformer layer corresponds to an optimization step, with every attention operation tied intuitively to terms in the underlying energy functional. WBPT thus unifies variational analysis with deep learning, embedding interpretability into the model without compromising performance.

**Contributions.**

- *Variational Perspective on Prompt-based Restoration:* Guided restoration is formulated as a tailored *STV* problem. Its optimization dynamics reveal a white-box attention mechanism, offering a principled explanation of how prompts influence Transformer attention.

- *White-Box Prompt Transformer:* A cascaded Transformer derived by unrolling the *STV* gradient flow, where each layer corresponds to an optimization step and attention operations align with terms in the underlying energy functional.

- *Bridging Classical and Modern AI:* The framework connects variational principles with deep prompt-based models, providing a foundation for interpretable image restoration and controllable attention mechanisms with clear theoretical grounding.

- *Empirical Validation:* WBPT achieves state-of-the-art results on multiple image restoration benchmarks while enabling transparent analysis of prompt–attention dynamics via rigorous, comprehensive visualization and controlled perturbation studies.

## 2 METHODS

In this section, we introduce WBPT, a variationally inspired framework for guided image restoration that interprets the guidance modality as *soft prompts* in a principled manner. WBPT integrates a tailored *STV* prior with learnable transformations $\mathbf{W}_i$ and soft prompt tokens $\boldsymbol{\xi}_i$, achieving restoration by unrolling $K$ gradient-flow steps. The overall pipeline and information flow across the $K$ cascaded stages are illustrated in Fig. 1, as depicted schematically.

## 2.1 Structured Modeling Framework for Guided Image Restoration

In guided image restoration, structural consistency across image modalities—where one modality (e.g., modality $A$) provides complementary information to enhance the restoration of another modality (e.g., modality $B$)—can be effectively exploited in practice. Rapidly acquired or higher-quality modality $A$ images can serve as informative priors to guide the restoration of modality $B$ (Ehrhardt & Betcke, 2016b). To systematically and rigorously model this guidance, we treat the modality $A$ image as a *prompt*, explicitly encoding its anatomical information through dedicated operators to assist in restoring modality $B$ (Potlapalli et al., 2023; Jia et al., 2022).

Formally, guided image restoration can be expressed as the following optimization problem:

$$\min_{\mathbf{x} \in \mathcal{X}} \mathcal{R}(\mathbf{x}; \boldsymbol{\xi}) := \mathcal{R}_1(\mathbf{x}) - \mu \mathcal{R}_2(\mathbf{x}; \boldsymbol{\xi}), \tag{1}$$

where $\mathbf{x} : \Omega \to \mathbb{R}^d$ denotes the target image (modality $B$) to be restored, and $\boldsymbol{\xi}$ represents the feature embedding of the guidance image (modality $A$). The function space $\mathcal{X}$ is chosen appropriately (e.g., Sobolev space $H^1(\Omega; \mathbb{R}^d)$ or $L^2(\Omega; \mathbb{R}^d)$) to ensure well-definedness of the optimization and its variational derivatives (Evans, 2022). The functional $\mathcal{R}_1(\mathbf{x})$ encodes intrinsic priors of $\mathbf{x}$, while minimizing $-\mathcal{R}_2(\mathbf{x}; \boldsymbol{\xi})$ enforces consistency between $\mathbf{x}$ and $\boldsymbol{\xi}$ in the transformed domain. The parameter $\mu > 0$ balances this trade-off.

Specifically, we redesign an enhanced *STV* prior to represent $\mathcal{R}$, which not only characterizes the structural priors of the target image but also enforces consistency with guidance features $\boldsymbol{\xi}_i$ in the domain defined by $\mathbf{W}_i$. This motivates our proposed weighted prompt formulation:

$$\mathcal{R}\left(\mathbf{x}; \{\mathbf{W}_i\}_{i=1}^N, \{\boldsymbol{\xi}_i\}_{i=1}^N\right) := \frac{1}{2} \sum_{i=1}^N \int_\Omega \mathrm{Tr}\big(\psi\big(\mathbf{W}_i \mathbf{x}(s)(\mathbf{W}_i \mathbf{x}(s))^\top\big)\big)\, ds$$
$$- \mu \sum_{i=1}^N \int_\Omega \mathrm{Tr}\big(\psi\big(\boldsymbol{\xi}_i(s)(\mathbf{W}_i \mathbf{x}(s))^\top\big)\big)\, ds. \tag{2}$$

Classical *STV* instantiates $\mathbf{W}_i$ as gradient operators capturing local structures, whereas nonlocal *STV* incorporates global interactions for superior performance. Motivated by this, we parameterize $\mathbf{W}_i$ as learnable global transformations via fully connected layers rather than local convolutional kernels (Wang et al., 2018). In parallel, $\{\boldsymbol{\xi}_i\}_{i=1}^N$ serve as prompts in the transformed domain. When explicit guidance images are unavailable, these prompts are relaxed into learnable soft tokens (Jia et al., 2022; Potlapalli et al., 2023). Finally, $\psi(\cdot)$ is a sparsity-inducing penalty, for which nonconvex forms such as $\psi(u) = \ln(1 + u)$ are effective.

## 2.2 White-box Prompt Transformer via Variational Derivation

The gradient of the energy functional (2) is derived via variational calculus, resulting in an interpretable form:

$$\frac{\delta \mathcal{R}(\mathbf{x}; \mathbf{W}_i, \boldsymbol{\xi}_i)}{\delta \mathbf{x}} \approx \mathbf{W}_i^* \mathbf{W}_i \mathbf{x} + \gamma \mathbf{W}_i^* \mathbf{W}_i \mathbf{x} \cdot \mathrm{softmax}\big((\mathbf{W}_i \mathbf{x})^\top \mathbf{W}_i \mathbf{x}\big)$$
$$- \mu\, \mathbf{W}_i^* \boldsymbol{\xi}_i - \gamma \mu\, \mathbf{W}_i^* \boldsymbol{\xi}_i \cdot \mathrm{softmax}\big((\mathbf{W}_i \mathbf{x})^\top \boldsymbol{\xi}_i\big). \tag{3}$$

Here, $\mathcal{R}(\mathbf{x}; \mathbf{W}_i, \boldsymbol{\xi}_i)$ denotes the $i$-th component of

$$\mathcal{R}\big(\mathbf{x}; \{\mathbf{W}_i\}_{i=1}^N, \{\boldsymbol{\xi}_i\}_{i=1}^N\big) = \sum_{i=1}^N \mathcal{R}(\mathbf{x}; \mathbf{W}_i, \boldsymbol{\xi}_i).$$

Although Eq.3 replaces linear inner-product weights by column-wise softmax weights, Appendix C shows that, after optimal column-wise scaling, the resulting attention-based form admits a closed-form error expression with an explicit condition under which this approximation is highly accurate.

This gradient inspires the *Single-Prompted Structure Attention (SPSA)* module:

$$\mathrm{SPSA}(\mathbf{x} \mid \mathbf{W}_i, \boldsymbol{\xi}_i, \gamma, \mu) := \mathbf{W}_i \mathbf{x} \cdot \mathrm{softmax}\big((\mathbf{W}_i \mathbf{x})^\top \mathbf{W}_i \mathbf{x}\big) - \gamma \mu\, \boldsymbol{\xi}_i \cdot \mathrm{softmax}\big((\mathbf{W}_i \mathbf{x})^\top \boldsymbol{\xi}_i\big) + \mu\, \boldsymbol{\xi}_i. \tag{4}$$

Eq.4 can thus be viewed as an operator-level implementation of this gradient-to-attention mapping, and inherits the same error bound and validity condition.

For multiple prompts, we define the *Multi-Prompted Structure Attention (MPSA)* module:

$$\text{MPSA}(\mathbf{x} \mid \{\mathbf{W}_i\}, \{\boldsymbol{\xi}_i\}, \gamma, \mu) := [\mathbf{W}_1^* \quad \cdots \quad \mathbf{W}_N^*] \begin{bmatrix} \text{SPSA}(\mathbf{x} \mid \mathbf{W}_1, \boldsymbol{\xi}_1, \gamma, \mu) \\ \vdots \\ \text{SPSA}(\mathbf{x} \mid \mathbf{W}_N, \boldsymbol{\xi}_N, \gamma, \mu) \end{bmatrix}. \tag{5}$$

This formulation provides an explicitly controllable, prompt-driven white-box attention mechanism with three functional components:

- *Self-Reconstruction Term:* $\mathbf{W}_i\mathbf{x} \cdot \text{softmax}((\mathbf{W}_i\mathbf{x})^\top \mathbf{W}_i\mathbf{x})$, enhancing intrinsic feature coherence via self-expression.
- *Prompt-Alignment Term:* $-\lambda_1\boldsymbol{\xi}_i \cdot \text{softmax}((\mathbf{W}_i\mathbf{x})^\top \boldsymbol{\xi}_i)$, introducing a repulsive force to prevent trivial imitation while enabling structural adaptation.
- *Prompt-Bias Term:* $+\lambda_2\boldsymbol{\xi}_i$, injecting prior knowledge as a static inductive bias to ensure faithful restoration.

### 2.3 Cascaded Transformer Architecture via Gradient Flow Unrolling

To optimize (2) while enforcing consistency with measurements $\mathbf{y}$, we consider the continuous-time gradient flow:

$$\frac{\partial \mathbf{x}(t)}{\partial t} = -\left(\frac{\delta\mathcal{R}}{\delta\mathbf{x}}(\mathbf{x}(t)) + \Delta(\mathbf{x}(t), \mathbf{y})\right), \tag{6}$$

where $\Delta(\mathbf{x}(t), \mathbf{y})$ denotes the gradient of the data fidelity term.

Discretizing via explicit Euler with step size $\eta$ gives:

$$\mathbf{x}_{k+1} = \mathbf{x}_k - \eta\left(\frac{\delta\mathcal{R}}{\delta\mathbf{x}}(\mathbf{x}_k) + \Delta(\mathbf{x}_k, \mathbf{y})\right), \tag{7}$$

where $k$ indexes the iteration. Substituting the MPSA module (5), the update becomes:

$$\mathbf{x}_{k+1} = \left(\boldsymbol{I} - \eta\sum_{i=1}^{N}\mathbf{W}_i^*\mathbf{W}_i\right)\mathbf{x}_k - \eta\text{MPSA}\left(\mathbf{x}_k \mid \{\mathbf{W}_i\}, \{\boldsymbol{\xi}_i\}, \gamma, \mu\right) - \eta\Delta_\phi(\mathbf{x}_k, \mathbf{y}). \tag{8}$$

In practice, the forward degradation model is often unknown, so $\Delta$ cannot be computed explicitly. We replace it with a learnable data-consistency term $\Delta_\phi(\mathbf{x}_k, \mathbf{y})$, parameterized by $\phi$. This module captures the discrepancy between $\mathbf{x}_k$ and $\mathbf{y}$ while interacting with $\{\boldsymbol{\xi}_i\}$ and $\{\mathbf{W}_i\}$.

Unrolling $K$ iterations of this process yields a deep cascaded network alternating between prompt-driven attention blocks and learnable data-consistency modules.

## 3 Experiments

We evaluate our WBPT against both general-purpose restoration methods and specialized All-in-One approaches (Table 1). Averaged across tasks, WBPT raises the mean PSNR from 30.16 to 31.02,dB, narrowing the gap to PromptIR while maintaining transparency and controllability.

To explicitly assess the effect of multi-scale processing, we further introduce WBPT[†] as a "plus" variant: it augments the fully white-box single-scale WBPT with an additional pyramid pathway. Within this pathway, the feature interactions are still governed by our STV-consistent white-box attention blocks, while the down-/up-sampling operators for cross-scale aggregation are implemented as a learned, black-box module. Thus, WBPT[†] retains a white-box core but uses a partially black-box multi-scale extension. A fully differentiable white-box pyramid is an active direction of ongoing work. Empirically, WBPT[†] matches PromptIR on average and exceeds it on selected tasks.

Task-level results further highlight the advantages of the proposed framework. On Rain100L (deraining), WBPT[†] surpasses PromptIR, and Fig. 2 confirms removal of rain streaks with diverse

Table 1: Comparison in the All-in-One restoration setting. Results are reported as PSNR/SSIM. Within each block (single-scale vs. multi-scale), the best and second-best are **boldfaced** and underlined, and gray shading indicates models whose core attention and reconstruction dynamics are white-box. WBPT[†] shares the same white-box core as WBPT but incorporates a learned, black-box pyramid pathway for multi-scale aggregation.. WBPF yields a marked improvement over WBT, while WBPF[†] achieves performance comparable to PromptIR and surpasses it on several tasks.

| Method | Dehazing SOTS | Deraining Rain100L | Denoising (BSD68) | | | Avg. |
| | | | $\sigma$=15 | $\sigma$=25 | $\sigma$=50 | |
| --- | --- | --- | --- | --- | --- | --- |
| *Single-scale methods* | | | | | | |
| BRDNet | 23.23/0.895 | 27.42/0.895 | 32.26/0.898 | 29.76/0.836 | 26.34/**0.836** | 27.80/0.843 |
| FDGAN | 24.71/0.924 | 29.89/0.933 | 30.25/0.910 | 28.81/0.868 | 26.43/0.776 | 28.02/0.883 |
| AirNet | 27.94/0.962 | 34.90/0.967 | **33.92/0.933** | **31.26/0.888** | **28.00**/0.797 | **31.20/0.910** |
| WBT | 27.40/0.958 | 32.13/0.940 | 33.17/0.923 | 30.68/0.875 | 27.41/0.770 | 30.16/0.893 |
| WBPT | **29.31/0.972** | **35.93/0.971** | 33.66/0.929 | 31.01/0.881 | 27.72/0.781 | 31.02/0.907 |
| *Multi-scale methods* | | | | | | |
| LPNet | 20.84/0.828 | 24.88/0.784 | 26.47/0.778 | 24.77/0.748 | 21.26/0.552 | 23.64/0.738 |
| MPRNet | 25.28/0.954 | 33.57/0.954 | 33.54/0.927 | 30.89/0.880 | 27.56/0.779 | 30.17/0.899 |
| DL | 26.92/0.391 | 32.62/0.931 | 33.05/0.914 | 30.41/0.861 | 26.90/0.740 | 29.98/ 0.875 |
| PromptIR | **30.58/0.974** | 36.37/0.972 | **33.98**/0.933 | **31.31**/0.888 | 28.06/0.799 | **32.06**/0.913 |
| WBPT[†] | 29.94/0.970 | **37.08/0.974** | 33.86/**0.934** | 31.28/**0.890** | **28.08/0.801** | 32.05/**0.914** |

orientations more thoroughly than WBPT, producing cleaner outputs. On SOTS (dehazing), while WBPT[†] falls short of PromptIR in PSNR, it demonstrates the benefit of multi-scale modeling; as shown in Fig. 3, haze removal is clearer and scene details are better preserved. On BSD68 (denoising), WBPT[†] achieves PSNR comparable to PromptIR and yields consistently higher SSIM.

**Datasets.** For denoising, training is conducted on BSD400 (Arbelaez et al., 2010) and WED (Ma et al., 2016) with Gaussian noise levels $\sigma \in \{15, 25, 50\}$, and evaluation is performed on BSD68 (Martin et al., 2001) and Urban100 (Huang et al., 2015). For deraining, we use Rain100L (200 training / 100 test images) (Yang et al., 2020). For dehazing, training is on SOTS (72,135 images) and evaluation on SOTS (500 images) (Li et al., 2018). In the All-in-One setting, these datasets are combined to train a unified model, following the protocol of (Potlapalli et al., 2023).

**Model and Training.** WBPT is trained end-to-end using a 10-iteration white-box framework, where each iteration integrates a learnable gradient update with a Transformer-based prompt branch. Prompts are injected specifically at the sixth Transformer block in each iteration. Training is performed with the standard Adam optimizer ($\beta_1$=0.9, $\beta_2$=0.999) at a fixed learning rate of $1 \times 10^{-4}$ for 120 epochs in total. We use random $128 \times 128$ crops with rotations and flips, optimizing with L2 (MSE) loss. The best checkpoint is selected based on validation performance.

### 3.1 MULTIPLE DEGRADATION ALL-IN-ONE RESULTS

We compare our white-box models with general-purpose restoration approaches and specialized All-in-One methods (Table 1). Averaged across tasks, WBPT raises the mean PSNR from 30.16 to 31.02 dB, narrowing the gap to PromptIR while preserving transparency and controllability. Moreover, our multi-scale extension WBPT[†], which augments the white-box WBPT core with a learned pyramid aggregator, performs on par with PromptIR and even surpasses it on certain tasks. On Rain100L for deraining, WBPT[†] outperforms PromptIR; visual comparisons in Fig. 2 show that, relative to WBT, WBPT more effectively removes rain streaks of diverse orientations, yielding cleaner rain-free results. On SOTS for dehazing, although WBPT[†] does not surpass PromptIR, it confirms the benefits of the multi-scale design; examples in Fig. 3 indicate clearer haze removal and more faithful scene restoration. On BSD68 for denoising, WBPT[†] attains PSNR comparable to PromptIR while overall delivering higher SSIM.

### 3.2 SINGLE DEGRADATION ONE-BY-ONE RESULTS

We evaluate PromptIR under the single-task setting, where a separate model is trained for each restoration task. This setting is intended to empirically verify that content-adaptive prompting via the prompt block is also effective for single-task networks. Table 2 reports the deraining results on standard datasets: our single-scale white-box WBPT consistently achieves the best performance,

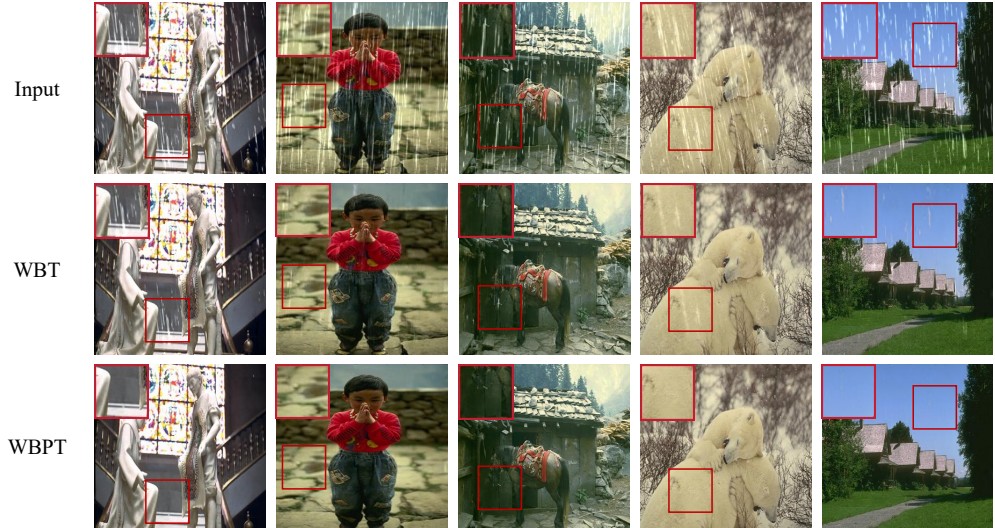

Figure 2: Deraining results under the All-in-One setting. Compared with WBT, our WBPT removes numerous residual rain streaks that WBT fails to eliminate, yielding cleaner backgrounds and sharper details (see red zoom-in boxes).

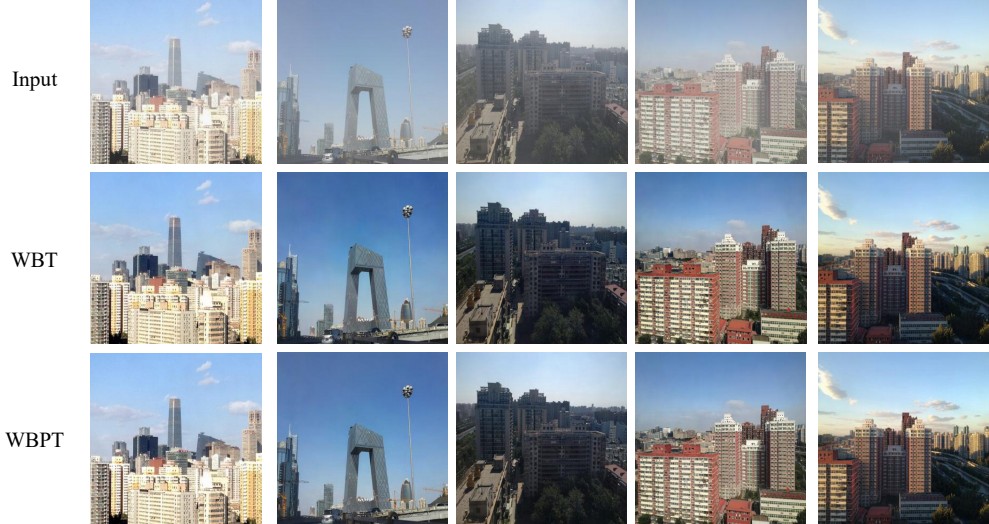

Figure 3: Dehazing results for all-in-one methods.Compared with WBT, our WBPT recovers clearer sky regions and sharper building edges, suppresses veiling glare and color cast, and yields more natural contrast and details across diverse urban scenes.

surpassing PromptIR and the multi-scale extension WBPT$^{\dagger}$ with a learned pyramid aggregator; relative to WBT (without prompts), WBPT delivers a 1.93 dB gain in PSNR. For dehazing and denoising, although WBPT$^{\dagger}$ does not surpass PromptIR, it achieves comparable performance; see Tables 3 and 4.

## 3.3 ATTENTION VISUALIZATION

To examine differences in model focus, we visualize the last-layer multi-head attention of the white-box reconstruction model (WBPT) and the black-box method PromptIR. For each model, we extract the last-layer attention tensor $A \in \mathbb{R}^{H \times N \times N}$, average across heads, and further aggregate along the query dimension to obtain a single-channel response for each window. The window-wise responses are then reassembled into a full-image heatmap via reverse window stitching. To ensure a fair comparison, both models use identical inputs and visualization settings. As shown in Fig. 5, WBPT exhibits strong responses on object boundaries and structural regions, indicating a preference for image geometry and semantic content rather than directly following degradation textures; notably, this boundary-centric attention aligns with our STV objective (Sec. 2.1). In contrast, PromptIR's

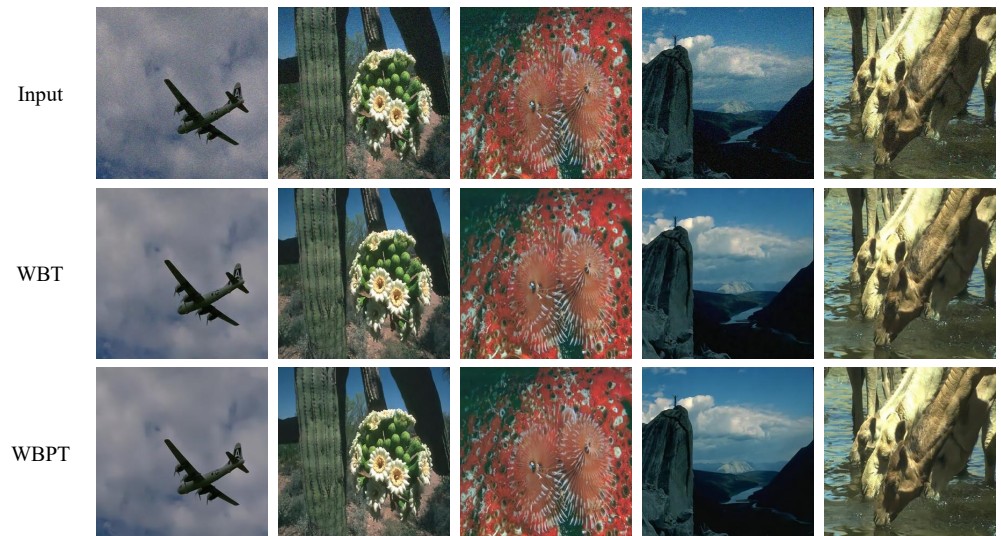

Figure 4: Denoising results for all-in-one methods.

Table 2: Deraining results on Rain100L in the single-task setting. Within each scale group (Single or Multi), best results are **boldfaced** and second-best are underlined; Gray indicates white-box models.

| Scale | Single-scale methods | | | | Multi-scale methods | | | | |
|---|---|---|---|---|---|---|---|---|---|
| Method | SIRR | AirNet | WBT | WBPT | MSPFN | LPNet | Restormer | PromptIR | WBPT[†] |
| PSNR | 32.37 | 34.90 | 36.77 | **38.70** | 33.50 | 33.61 | 36.74 | 37.04 | **38.54** |
| SSIM | 0.926 | 0.977 | 0.977 | **0.983** | 0.948 | 0.958 | 0.978 | 0.979 | **0.984** |

responses align with rain streaks and are more tightly coupled to the degradation pattern, suggesting a greater reliance on degradation-pattern detection.

### 3.4 t-SNE Analysis of Intermediate Representations

To analyze the intermediate representations of an all-in-one model across different degradations, we tap the *input* to the final convolution layer of the Transformer backbone during the forward pass. For each image, we apply global average pooling over the spatial dimensions to obtain a channel-wise embedding vector. We collect embeddings from three standard test sets—BSD68 denoising ($\sigma = 25$), Rain100L deraining, and SOTS-Outdoor dehazing—and project them to 2D using t-SNE.

Figure 6 compares the black-box PromptIR with our white-box WBPT under identical preprocessing and t-SNE settings: PromptIR (left) yields highly entangled embeddings with substantial cross-task overlap, whereas WBPT (right) forms well-separated clusters for the Noisy, Hazy, and Rainy samples, exhibiting tighter intra-cluster compactness and clearer inter-cluster margins. These results indicate that WBPT learns more discriminative, task-aware representations in the all-in-one setting.

To further examine whether the learned representations remain degradation-aware beyond single degradations, we additionally conduct t-SNE analysis on the CDD11 mixed-degradation dataset. CDD11 contains several two-way combinations of degradations; we select three representative protocols (low+haze, low+snow, haze+snow) and, following exactly the same feature-extraction protocol as above, compute embeddings for PromptIR and WBPT (Figure 7). While the three protocols share degradation components pairwise, the PromptIR embeddings still exhibit substantial cross-class overlap, whereas WBPT forms more compact clusters with clearer boundaries between mixed degradations. This suggests that our white-box design continues to organize intermediate features primarily according to degradation combinations, rather than image content, even in more complex mixed-degradation scenarios.

### 3.5 Stability under prompt-parameter perturbations

To verify the stability of WBPT under prompt-parameter perturbations, we conduct a perturbation-sensitivity study in a controlled experimental setting and compare it with PromptIR. The test datasets are BSD68 (denoising), Rain100L (deraining), and SOTS-Outdoor (dehazing). We inject additive Gaussian noise *only* into the prompt parameters ($\sigma \in [0.001, 0.1]$), while keeping all other settings

Table 3: Dehazing results on SOTS dataset in the single-task setting. Within each block (single-scale vs. multi-scale), the best and second-best are **boldfaced** and underlined, and gray shading indicates white-box models. PSNR/SSIM reported; higher is better.

| Scale | | *Single-scale methods* | | | | | *Multi-scale methods* | | |
|---|---|---|---|---|---|---|---|---|---|
| Method | AODNet | FDGAN | AirNet | WBT | WBPT | EPDN | Restormer | PromptIR | WBPT$^\dagger$ |
| PSNR | 20.29 | 23.15 | 23.18 | **28.72** | 28.33 | 22.57 | 30.87 | **31.31** | 30.47 |
| SSIM | 0.877 | 0.921 | 0.900 | 0.961 | **0.967** | 0.863 | 0.969 | **0.973** | 0.972 |

Table 4: Denoising comparisons in the single-task setting on BSD68 and Urban100. Results are reported as PSNR/SSIM. Within each block (single-scale vs. multi-scale), the best and second-best are **boldfaced** and underlined, respectively. gray shading indicates white-box models. At the challenging noise level of $\sigma = 50$ on Urban100, our WBPT achieves a 0.39 dB improvement over WBT. Meanwhile, WBPT$^\dagger$ attains performance comparable to PromptIR.

| Method | BSD68 | | | Urban100 | | |
|---|---|---|---|---|---|---|
| | $\sigma$=15 | $\sigma$=25 | $\sigma$=50 | $\sigma$=15 | $\sigma$=25 | $\sigma$=50 |
| *Single-scale methods* | | | | | | |
| CBM3D | 33.50/0.922 | 30.69/0.868 | 27.36/0.763 | 33.93/0.941 | 31.36/0.909 | 27.93/0.840 |
| DnCNN | 33.89/0.930 | 31.23/0.883 | 27.92/0.789 | 32.98/0.931 | 30.81/0.902 | 27.59/0.833 |
| IRCNN | 33.87/0.929 | 31.18/0.882 | 27.88/0.790 | 27.59/0.833 | 31.20/0.909 | 27.70/0.840 |
| FFDNet | 33.87/0.929 | 31.21/0.882 | 27.96/0.789 | 33.83/0.942 | 31.40/0.912 | 28.05/0.848 |
| BRDNet | 34.10/0.929 | 31.43/0.885 | 28.16/0.794 | **34.42**/0.946 | 31.99/0.919 | 28.56/0.858 |
| AirNet | **34.14/0.936** | **31.48/0.893** | **28.23/0.806** | 34.40/0.949 | **32.10**/0.924 | **28.88/0.871** |
| WBT | 33.59/0.930 | 30.92/0.882 | 27.85/0.793 | 33.43/0.956 | 30.42/0.924 | 27.18/0.858 |
| WBPT | 34.02/0.935 | 31.35/0.891 | 28.03/0.797 | 34.15/**0.963** | 31.38/**0.937** | 27.57/0.870 |
| *Multi-scale methods* | | | | | | |
| Restormer | 34.29/0.937 | 31.64/0.895 | 28.41/0.810 | 34.67/**0.969** | 32.41/0.927 | 29.31/0.878 |
| PromptIR | **34.34**/0.938 | **31.71/0.897** | **28.49/0.813** | **34.77**/0.952 | **32.49/0.929** | **29.39/0.881** |
| WBPT$^\dagger$ | 34.31/**0.938** | 31.60/0.895 | 28.36/0.811 | 34.76/0.952 | 32.27/0.927 | 29.08/0.877 |

Table 5: Evaluation of stability under prompt-parameter perturbations, reported as relative drops in PSNR and SSIM (lower is better). Gaussian noise with $\sigma \in [0.001, 0.1]$ is injected exclusively into the prompt parameters. Results are averaged over multiple severity levels on BSD68, Rain100L, and SOTS-Outdoor. WBPT exhibits smaller drops than PromptIR, indicating greater stability.

| Model | Denoising | | Deraining | | Dehazing | | Average | |
|---|---|---|---|---|---|---|---|---|
| | PSNR | SSIM | PSNR | SSIM | PSNR | SSIM | PSNR | SSIM |
| PromptIR | -10.2% | -13.0% | -13.8% | -12.4% | -13.0% | -12.2% | -12.3% | -12.5% |
| WBPT | -3.05% | -0.35% | -1.02% | -0.64% | -2.18% | -0.05% | -2.08% | -0.31% |

(e.g., the prompt insertion layer) identical to the previous configuration. The evaluation metric is the *average performance drop percentage* (lower indicates higher stability), averaged over multiple perturbation severities and the three datasets. The corresponding averages are summarized in Table 5; a representative qualitative comparison at $\sigma = 0.1$ is shown in Fig. 8.

From the visual results, PromptIR consistently exhibits *systematic contrast and color shifts* after perturbing the prompt, suggesting an undesirable coupling between the prompt representation and global imaging attributes. Under prompt-only perturbations, such global tone/contrast changes are not what degradation awareness is expected to primarily induce. In contrast, WBPT with $\sigma = 0.1$ still removes rain effectively while maintaining remarkably stable contrast and colors, indicating stronger robustness and better degradation–prompt decoupling.

## 3.6 Guidance Modality Validation: Soft Prompt vs Real Guidance

We compare a learnable *soft prompt* with a proxy of real guidance (*hard*: image gradients → edge map plus Gaussian noise, $\sigma \in \{0.01, 0.02\}$). To control compute and isolate the modality effect, the *backbone is frozen* and only the prompt and fusion parameters are finetuned. For each test image, we report the paired difference $\Delta = \text{metric}_{\text{soft}} - \text{metric}_{\text{hard}}$; our goal is to assess the *relative* gap be-

PromptIR

WBPT

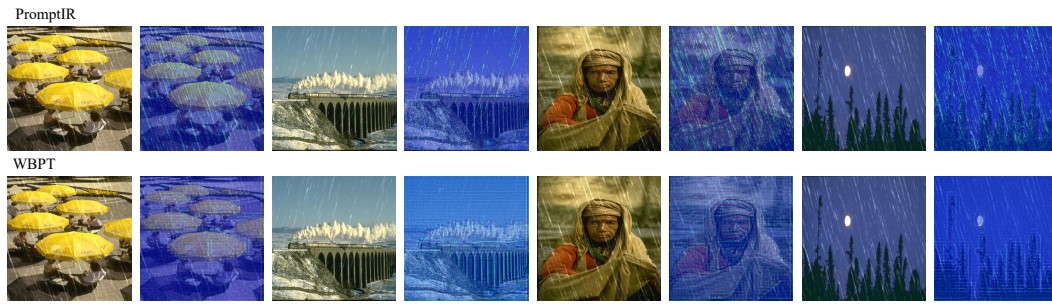

Figure 5: Attention-map visualizations for WBPT and PromptIR. Odd-numbered columns show input images; even-numbered columns show the corresponding attention maps. Top row: PromptIR; bottom row: WBPT. Attention heads and queries from the final layer are aggregated, and full-image heatmaps are reconstructed via reverse window stitching. WBPT focuses on object boundaries and main structures, whereas PromptIR emphasizes rain streaks.

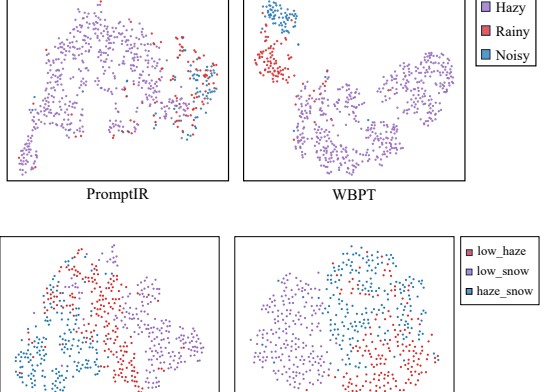

PromptIR          WBPT

- Hazy
- Rainy
- Noisy

Figure 6: t-SNE visualization of degradation embeddings from PromptIR and WBPT. Colors denote degradation types. With identical inputs and t-SNE settings, WBPT yields clearly separable clusters by degradation type, whereas PromptIR is more entangled.

PromptIR          WBPT

- low_haze
- low_snow
- haze_snow

Figure 7: t-SNE visualization of mixed-degradation embeddings on CDD11. We follow the same t-SNE settings as Figure 6. Colors indicate three mixed-degradation protocols (low+haze, low+snow, haze+snow) for PromptIR (left) and WBPT (right).

tween soft and hard rather than absolute gains. Equivalence margins are pre-registered as $\pm 0.02$ dB (PSNR) and $\pm 0.002$ (SSIM), within which soft and hard are deemed practically equivalent.

Under the finetune-only setting (backbone frozen), soft and hard guidance behave nearly identically across denoising, deraining, and dehazing in our controlled evaluations. The paired differences $\Delta = \text{soft} - \text{hard}$ are consistently tiny and remain within the pre-registered equivalence margins ($\pm 0.02$,dB PSNR / $\pm 0.002$ SSIM) for both $\sigma = 0.01$ and $\sigma = 0.02$. While dehazing yields lower absolute scores—reflecting its higher difficulty—the relative gap between soft and hard stays stable, which is precisely the comparison this experiment aims to isolate.

Table 6: Soft vs. hard guidance under the finetune-only setting (backbone frozen). Metrics are PSNR and SSIM in separate columns; parentheses denote the change relative to the baseline in the same column. Gray denote the paired difference $\Delta = \text{soft} - \text{hard}$; values within the pre-registered equivalence margins ($\pm 0.10$ dB PSNR, $\pm 0.002$ SSIM) indicate practical equivalence. Results are reported for $\sigma \in \{0.01, 0.02\}$. While absolute scores for dehazing are lower due to its higher difficulty, the soft–hard gap remains small and stable across $\sigma$.

| $\sigma$ | Task | Denoise | | Derain | | Dehaze | |
|---|---|---|---|---|---|---|---|
| | | PSNR | SSIM | PSNR | SSIM | PSNR | SSIM |
| 0.01 | WBT | 33.59 | 0.930 | 36.77 | 0.977 | 28.72 | 0.961 |
| | WBT+soft | 33.61 (+0.02) | 0.929 (−0.1%) | 36.90 (+0.13) | 0.978 (+0.1%) | 27.83 (−0.89) | 0.956 (−0.5%) |
| | WBT+hard | 33.63 (+0.04) | 0.929 (−0.1%) | 36.89 (+0.12) | 0.978 (+0.1%) | 27.87 (−0.85) | 0.958 (−0.3%) |
| | $\Delta$ (soft−hard) | −0.02 | 0.000 | +0.01 | 0.000 | −0.04 | −0.002 |
| 0.02 | WBT | 33.59 | 0.930 | 36.77 | 0.977 | 28.72 | 0.961 |
| | WBT+soft | 33.62 (+0.03) | 0.930 (+0.0%) | 36.89 (+0.12) | 0.978 (+0.1%) | 27.83 (−0.89) | 0.956 (−0.5%) |
| | WBT+hard | 33.60 (+0.01) | 0.929 (−0.1%) | 36.89 (+0.12) | 0.978 (+0.1%) | 27.86 (−0.86) | 0.958 (−0.3%) |
| | $\Delta$ (soft−hard) | +0.02 | 0.000 | +0.00 | 0.000 | −0.03 | −0.002 |

PromptIR

WBPT

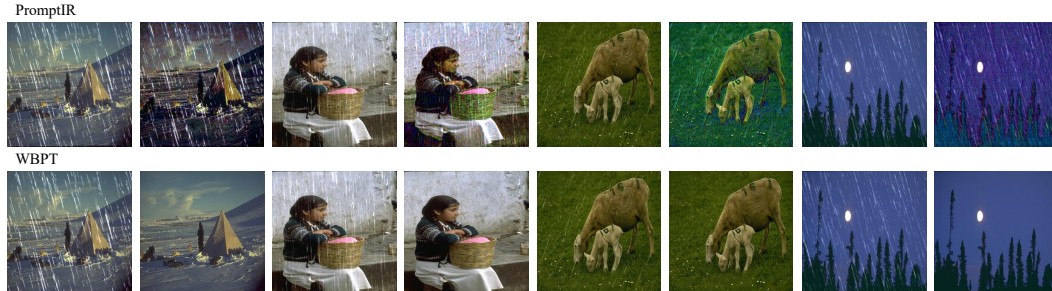

Figure 8: Qualitative comparison under Gaussian perturbation of prompt parameters ($\sigma = 0.1$). Odd-numbered columns show input images; even-numbered columns show restored results. Top row: PromptIR; bottom row: WBPT. WBPT preserves deraining quality, contrast, and colors, whereas PromptIR exhibits noticeable shifts, indicating lower robustness.

## 4 CONCLUSION

In this work, we revisited prompt-based Transformers from a variational perspective and established a principled connection between prompts and attention. By casting guided image restoration as a *STV* problem, we derived a white-box attention mechanism that offers an interpretable foundation for prompt–attention coupling. Building on this formulation, we unrolled the gradient flow into WBPT, a cascaded architecture that integrates variational principles with modern prompt learning. Extensive experiments on the classic three-degradation benchmark (denoising, deraining, dehazing) demonstrate that WBPT delivers competitive performance while maintaining transparent and robust prompt–attention dynamics. Beyond empirical performance, WBPT is complementary to recent all-in-one restorers such as Perceive-IR, DA-RCOT, MoCE-IR, AdaIR and DFPIR, which push state-of-the-art results via stronger backbones and degradation-aware modules. Our STV-based attention block can serve as an interpretable replacement or constraint for prompt/attention submodules inside these powerful architectures, or be combined with their semantic-, frequency- and feature-perturbation designs, suggesting "white-box mechanism + strong backbone" hybrids as a promising direction for future all-in-one restoration. Extending WBPT from three-degradation settings to more diverse protocols, such as five-degradation, mixed-degradation and real-world benchmarks, is an important next step toward fully exploiting this connection in practice.

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

# APPENDIX

## A  RELATED WORK

**Transformer-based image restoration.**    Transformer architectures have advanced image restoration by modeling long-range dependencies, with strong single-task systems such as SwinIR, Restormer, and successors (Chen et al., 2021; Liang et al., 2021; Zamir et al., 2022; Chen et al., 2022; Wang et al., 2022). To curb task specialization, all-in-one models handle multiple degradations with a unified backbone (e.g., AirNet (Li et al., 2022)). More recent all-in-one restoration works further expand this line by systematizing unified benchmarks and architectures, including a comprehensive AiOIR survey (Jiang et al., 2025) and degradation-aware backbones such as DA-RCOT (Tang et al., 2025), MoCE-IR (Zamfir et al., 2025), and AdaIR (Cui et al., 2025). We likewise pursue unified restoration but differ in how conditioning is defined and used within the network. Parallel to Transformer priors, diffusion-based restoration has recently shown competitive performance and broad applicability(Kawar et al., 2022; Chung et al., 2023; Saharia et al., 2022), and our formulation is complementary: it can inject structure-aware conditioning regardless of the underlying prior family.

**Prompt-based conditioning for restoration.**    Prompt-based Transformers inject task or condition cues (e.g., degradation type, maps, or control signals) into a shared backbone to enable multi-task

adaptation (Potlapalli et al., 2023; Li et al., 2022; Tian et al., 2024; Kong et al., 2025). On top of these designs, Perceive-IR (Zhang et al., 2025) and DFPIR (Tian et al., 2025) further improve all-in-one performance via quality-aware prompt learning and degradation-aware feature perturbations, while Defusion (Luo et al., 2025) leverages diffusion priors with degradation instructions for unified restoration. Prevailing designs treat prompts as black-box tokens or channel-wise modulations (e.g., visual prompt tuning/adapters) that are concatenated to features and learned end-to-end, which limits interpretability and spatial control(Jia et al., 2022). In contrast, we treat prompts as **structural priors** and derive structure-aware prompt attention from a variational formulation via **learnable regularization gradients (LRG)**, providing mechanistic interpretability and spatial controllability (Yu et al., 2023).

**Model-based deep learning and unrolled networks.** Model-based approaches explicitly couple data fidelity with learned priors through algorithm unrolling (e.g., ADMM-Net, MoDL, VarNet) (Sun et al., 2016; Aggarwal et al., 2019; Hammernik et al., 2018; Adler & Öktem, 2018). Plug-and-play and RED families further connect optimization with learned denoisers (Venkatakrishnan et al., 2013; Yang et al., 2022; Ulyanov et al., 2018; Kamilov et al., 2023). Our design follows this lineage: we unfold a variational objective, keep a physics-consistency step, and introduce a **learnable data-consistency (LDC)** module that complements the physics-consistency step to suppress residual artifacts. A key difference is that our conditioning instead arises from a regularizer-driven decomposition and directly parameterizes attention via LRG.

**Structure-guided and cross-modal reconstruction (MRI).** Guided reconstruction leverages side information to align the target with structures visible in a guide modality; in MRI, T1 can guide T2 reconstruction under multi-contrast or cross-modal priors (Ehrhardt & Betcke, 2016b; Feng et al., 2021; Yang et al., 2022; Zhou et al., 2024). This line builds upon classical physics-consistent MRI, including parallel imaging and compressed sensing(Pruessmann et al., 1999; Lustig et al., 2007), and deep cascades that interleave learned priors with data consistency(Schlemper et al., 2018). We reinterpret guidance as prompting within a white-box formulation: the prompt enters the variational gradient as a structure-aligned term that controls attention, linking classical guided reconstruction with modern prompt-based Transformers.

**Summary.** Compared to black-box prompt injection, our **White-Box Prompt Transformer (WBPT)** provides a variationally grounded route to structure-aware attention (via LRG) while maintaining faithful reconstruction through an LDC-augmented fidelity step, unifying multi-task restoration and guided MRI within the same unfolded architecture.

# B   ABLATION EXPERIMENT

## B.1   SPSA COMPONENT ABLATIONS

We ablate the **SPSA** operator defined in Eq. 4 by removing its last two terms: $A$ $(-\gamma\mu\,\boldsymbol{\xi}_i \cdot \mathrm{softmax}((\mathbf{W}_i\mathbf{x})^\top\boldsymbol{\xi}_i))$ and $B$ $(+\mu\,\boldsymbol{\xi}_i)$. All other implementation details, training protocol, and hyperparameters (including $\gamma, \mu$) follow the main setup. On BSD68 at $\sigma = 50$, removing either component degrades performance, while the full model (Eq. 4) attains the best average results, indicating that the two terms play complementary roles.

## B.2   POSITION OF PROMPT BLOCKS.

In the hierarchical decoder, we ablate where to inject the prompt blocks. Table 8 compares placing prompts at blocks 1&2, at block 6, and at all decoder blocks on the denoising task with $\sigma = 15$. While placing prompts at all blocks yields only marginal gains over block 6 (up to $0.04\,\mathrm{dB}$ in PSNR and $0.001$ in SSIM), it introduces nontrivial computational and latency overhead. We therefore adopt the single-block design at block 6 as the default, which closely matches the all-block variant while reducing wall-clock time and memory footprint.

### B.3 COMPLEMENTARITY BETWEEN PROMPT AND DATA-CONSISTENCY MODULES.

To avoid attributing the overall improvement to a single component, we conduct a systematic ablation on the deraining task, comparing four configurations: removing the Prompt (w/o Prompt), removing the data-consistency module (w/o DC), removing both components as the cascaded baseline (w/o Prompt & DC), and enabling both components (Prompt+DC). To ensure fairness, all other settings—the number of unrolled steps $K$, step size $\eta$, training schedule, and parameter budget—are kept identical across variants. As summarized in Table 9, relative to the baseline without both modules (w/o Prompt DC), introducing either module alone yields consistent gains in PSNR/SSIM; enabling both simultaneously (Prompt+DC) produces the largest improvement, confirming the strong complementarity between the structural prior provided by the Prompt and the observation-consistency constraint enforced by the DC module.

Table 7: Ablation study of **SPSA** components in Eq. 4 on BSD68 with $\sigma = 50$. The complete formulation (Eq. 4) achieves the best overall performance. Here, $A$ corresponds to $-\gamma\mu\,\boldsymbol{\xi}_i\cdot\mathrm{softmax}((\mathbf{W}_i\mathbf{x})^\top\boldsymbol{\xi}_i)$, and $B$ corresponds to $+\mu\,\boldsymbol{\xi}_i$.

| Variant | PSNR | SSIM |
|---|---|---|
| w/o A | 27.57 | 0.753 |
| w/o B | 27.97 | 0.798 |
| Full | 28.33 | 0.967 |

Table 8: Ablation of prompt-injection positions on BSD68 at $\sigma = 15$. Injecting at block 6 matches all-block injection while substantially reducing computation. Results are shown for blocks 1&2, block 6, and all blocks.

| Placement | PSNR | SSIM |
|---|---|---|
| block 6 | 34.02 | 0.963 |
| blocks 1&2 | 33.97 | 0.962 |
| all blocks | 34.06 | 0.964 |

Table 9: Ablation study of **Prompt** and **Data-Consistency (DC)** components within the cascaded Transformer unrolled from Eq. 8, evaluated on the Rain100L dataset.

| Variant | PSNR | SSIM |
|---|---|---|
| w/o DC&prompt | 36.77 | 0.977 |
| w/o DC | 37.07 | 0.978 |
| w/o prompt | 37.46 | 0.979 |
| Prompt&DC | 38.70 | 0.983 |

### B.4 ACCURACY−OVERHEAD TRADE-OFF FOR PROMPT DESIGN

To further quantify the efficiency of the prompt design, we plot parameter-overhead vs. PSNR curves on Rain100L under the single-task deraining setting. Figure 9 summarizes three sets of experiments: (a) different multi-insertion strategies, (b) the number of prompt tokens N with insertion fixed at the 6-th decoder block, and (c) the projector rank R, again with insertion fixed at the 6-th block. For all panels, the horizontal axis denotes the relative parameter overhead (%) with respect to the "No Prompt" baseline, and the vertical axis reports PSNR (dB). We also provide the exact numerical values in the accompanying appendix tables.

## C VARIATIONAL DERIVATION OF THE ENERGY FUNCTIONAL

For notational simplicity, the subscript $i$ is omitted throughout this section. To develop the optimization algorithm for the proposed model, we consider the variational derivative of the energy functional $\mathcal{R}(\mathbf{x})$, defined as:

$$
\mathcal{R}(\mathbf{x}) = \frac{1}{2}\int_\Omega \mathrm{Tr}\left(\ln\left(\mathbf{I} + (\mathbf{W}\mathbf{x}(s))(\mathbf{W}\mathbf{x}(s))^\top\right)\right) ds
$$
$$
- \mu\int_\Omega \mathrm{Tr}\left(\ln\left(\mathbf{I} + (\mathbf{W}\mathbf{x}(s))(\boldsymbol{\xi}(s))^\top\right)\right) ds
$$

(9)

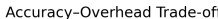

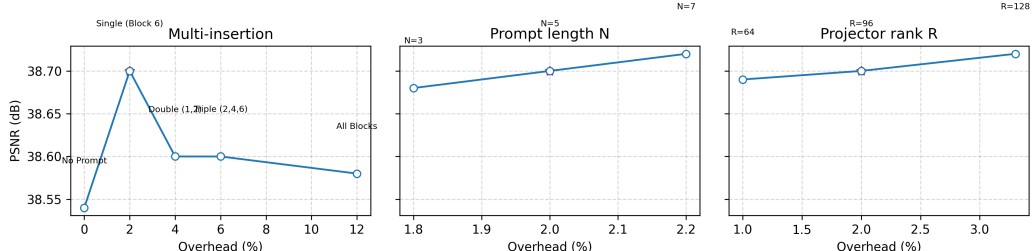

Figure 9: Accuracy–overhead trade-off for prompt design on Rain100L (single-task deraining). (a) Multi-insertion strategies (No Prompt, Single block 6, Double, Triple, All Blocks). (b) Number of prompt tokens N with insertion fixed at block 6. (c) Projector rank R of $W_i$ with insertion fixed at block 6. The horizontal axis shows parameter overhead (%) relative to the "No Prompt" baseline; the vertical axis reports PSNR (dB). Star markers indicate the default configuration used in the main paper (single insertion at block 6 with $N = 5$ and $R = 96$), which lies on or very close to the Pareto front of the accuracy–overhead trade-off.

where $\mathbf{x}(s)$ is the optimization variable, $\boldsymbol{\xi}(s)$ denotes the structural prompt, $\mathbf{W}$ is a linear transformation operator, and $\mu$ is a regularization weight.

Let us define $\mathbf{y}(s) = \mathbf{W}\mathbf{x}(s)$ and $\tilde{\mathbf{y}}(s) = \boldsymbol{\xi}(s)$.

## C.1 VARIATION OF THE FIRST TERM

Consider the first term:

$$\mathcal{R}_1(\mathbf{x}) = \frac{1}{2} \int_\Omega \mathrm{Tr}\left(\ln\left(\mathbf{I} + \mathbf{y}(s)\mathbf{y}(s)^\top\right)\right) ds \tag{10}$$

Using the matrix differential identity:

$$d\,\mathrm{Tr}\left(\ln\left(\mathbf{I} + \mathbf{y}\mathbf{y}^\top\right)\right) = \mathrm{Tr}\left(\left(\mathbf{I} + \mathbf{y}\mathbf{y}^\top\right)^{-1} d(\mathbf{y}\mathbf{y}^\top)\right) \tag{11}$$

Since $d(\mathbf{y}\mathbf{y}^\top) = d\mathbf{y} \cdot \mathbf{y}^\top + \mathbf{y} \cdot d\mathbf{y}^\top$, we have:

$$\begin{aligned}
d\,\mathrm{Tr}\left(\ln\left(\mathbf{I} + \mathbf{y}\mathbf{y}^\top\right)\right) &= \mathrm{Tr}\left(\left(\mathbf{I} + \mathbf{y}\mathbf{y}^\top\right)^{-1}\left(d\mathbf{y} \cdot \mathbf{y}^\top + \mathbf{y} \cdot d\mathbf{y}^\top\right)\right) \\
&= 2\mathrm{Tr}\left(\left(\mathbf{I} + \mathbf{y}\mathbf{y}^\top\right)^{-1}\mathbf{y}d\mathbf{y}^\top\right) \\
&= 2\left(\left(\mathbf{I} + \mathbf{y}\mathbf{y}^\top\right)^{-1}\mathbf{y}\right)^\top d\mathbf{y}
\end{aligned} \tag{12}$$

where the last equality follows from the identity $\mathrm{Tr}(A^\top) = \mathrm{Tr}(A)$ and $\mathrm{Tr}(A^\top B) = \langle A, B\rangle_F$.

Thus, the variation is:

$$d\,\mathrm{Tr}\left(\ln\left(\mathbf{I} + \mathbf{y}\mathbf{y}^\top\right)\right) = \left\langle 2\left(\mathbf{I} + \mathbf{y}\mathbf{y}^\top\right)^{-1}\mathbf{y}, d\mathbf{y}\right\rangle_F \tag{13}$$

Substituting $\mathbf{y}(s) = \mathbf{W}\mathbf{x}(s)$ and $d\mathbf{y}(s) = \mathbf{W}d\mathbf{x}(s)$, we obtain:

$$\begin{aligned}
d\mathcal{R}_1 &= \frac{1}{2} \int_\Omega \left\langle 2\left(\mathbf{I} + \mathbf{y}(s)\mathbf{y}(s)^\top\right)^{-1}\mathbf{y}(s), \mathbf{W}d\mathbf{x}(s)\right\rangle_F ds \\
&= \int_\Omega \left\langle \left(\mathbf{I} + \mathbf{y}(s)\mathbf{y}(s)^\top\right)^{-1}\mathbf{y}(s), \mathbf{W}d\mathbf{x}(s)\right\rangle_F ds
\end{aligned} \tag{14}$$

Using the definition of the adjoint operator $\mathbf{W}^*$:

$$dR_1 = \int_\Omega \left\langle \mathbf{W}^* \left( \left(\mathbf{I} + \mathbf{y}(s)\mathbf{y}(s)^\top\right)^{-1} \mathbf{y}(s)\right), d\mathbf{x}(s) \right\rangle_F ds \tag{15}$$

Therefore, the variational derivative is:

$$\frac{\delta R_1}{\delta \mathbf{x}}(s) = \mathbf{W}^* \left( \left(\mathbf{I} + \mathbf{y}(s)\mathbf{y}(s)^\top\right)^{-1} \mathbf{y}(s)\right) \tag{16}$$

## C.2 VARIATION OF THE SECOND TERM

Now consider the second term:

$$R_2(\mathbf{x}) = \int_\Omega \mathrm{Tr}\left(\ln\left(\mathbf{I} + \mathbf{y}(s)\tilde{\mathbf{y}}(s)^\top\right)\right) ds \tag{17}$$

Using the matrix differential identity:

$$d\,\mathrm{Tr}\left(\ln\left(\mathbf{I} + \mathbf{y}\tilde{\mathbf{y}}^\top\right)\right) = \mathrm{Tr}\left(\left(\mathbf{I} + \mathbf{y}\tilde{\mathbf{y}}^\top\right)^{-1} d(\mathbf{y}\tilde{\mathbf{y}}^\top)\right) \tag{18}$$

$$= \mathrm{Tr}\left(\left(\mathbf{I} + \mathbf{y}\tilde{\mathbf{y}}^\top\right)^{-1} d\mathbf{y}\tilde{\mathbf{y}}^\top\right) \tag{19}$$

where the second equality holds because $\tilde{\mathbf{y}}$ is independent of $\mathbf{x}$.

Using the cyclic property of the trace:

$$\mathrm{Tr}\left(\left(\mathbf{I} + \mathbf{y}\tilde{\mathbf{y}}^\top\right)^{-1} d\mathbf{y}\tilde{\mathbf{y}}^\top\right) = \mathrm{Tr}\left(\tilde{\mathbf{y}}^\top \left(\mathbf{I} + \mathbf{y}\tilde{\mathbf{y}}^\top\right)^{-1} d\mathbf{y}\right) \tag{20}$$

This can be written as an inner product:

$$d\,\mathrm{Tr}\left(\left(\mathbf{I} + \mathbf{y}\tilde{\mathbf{y}}^\top\right)^{-1} d\mathbf{y}\tilde{\mathbf{y}}^\top\right) = \left\langle \left(\mathbf{I} + \mathbf{y}\tilde{\mathbf{y}}^\top\right)^{-\top} \tilde{\mathbf{y}}, d\mathbf{y} \right\rangle_F \tag{21}$$

Substituting $\mathbf{y}(s) = \mathbf{W}\mathbf{x}(s)$ and $d\mathbf{y}(s) = \mathbf{W}d\mathbf{x}(s)$, we obtain:

$$dR_2 = \int_\Omega \left\langle \left(\mathbf{I} + \mathbf{y}(s)\tilde{\mathbf{y}}(s)^\top\right)^{-\top} \tilde{\mathbf{y}}(s), \mathbf{W}d\mathbf{x}(s) \right\rangle_F ds \tag{22}$$

Using the adjoint operator $\mathbf{W}^*$:

$$dR_2 = \int_\Omega \left\langle \mathbf{W}^* \left( \left(\mathbf{I} + \mathbf{y}(s)\tilde{\mathbf{y}}(s)^\top\right)^{-\top} \tilde{\mathbf{y}}(s)\right), d\mathbf{x}(s) \right\rangle_F ds \tag{23}$$

Therefore, the variational derivative is:

$$\frac{\delta R_2}{\delta \mathbf{x}}(s) = \mathbf{W}^* \left( \left(\mathbf{I} + \mathbf{y}(s)\tilde{\mathbf{y}}(s)^\top\right)^{-\top} \tilde{\mathbf{y}}(s)\right) \tag{24}$$

## C.3 COMBINING BOTH TERMS

Combining both components with their respective coefficients and regularization weight $\mu$, we derive the complete variational derivative:

$$\begin{aligned}
\frac{\delta R}{\delta \mathbf{x}}(s) &= \frac{\delta R_1}{\delta \mathbf{x}}(s) - \mu \frac{\delta R_2}{\delta \mathbf{x}}(s) \\
&= \mathbf{W}^* \left( \left(\mathbf{I} + \mathbf{y}(s)\mathbf{y}(s)^\top\right)^{-1} \mathbf{y}(s)\right) - \mu \mathbf{W}^* \left( \left(\mathbf{I} + \mathbf{y}(s)\tilde{\mathbf{y}}(s)^\top\right)^{-\top} \tilde{\mathbf{y}}(s)\right)
\end{aligned} \tag{25}$$

This gradient form supports the structure-aware optimization in the main algorithm and highlights the explicit interaction between the target variable $\mathbf{x}$ and the structural prompt $\boldsymbol{\xi}$ within the proposed framework.

## C.4 VARIATIONAL DERIVATIVE APPROXIMATION

Consider the variational derivative of the energy functional $\mathcal{R}[\mathbf{x}]$ with respect to $\mathbf{x}(s)$:

$$\frac{\delta\mathcal{R}}{\delta\mathbf{x}}(s) = \mathbf{W}^*\Big((\mathbf{I} + \mathbf{y}(s)\mathbf{y}(s)^\top)^{-1}\mathbf{y}(s)\Big) - \mu\mathbf{W}^*\Big((\mathbf{I} + \mathbf{y}(s)\tilde{\mathbf{y}}(s)^\top)^{-\top}\tilde{\mathbf{y}}(s)\Big), \tag{26}$$

where

$$\mathbf{y}(s) = \mathbf{W}\mathbf{x}(s), \qquad \tilde{\mathbf{y}}(s) = \boldsymbol{\xi}(s). \tag{27}$$

Using the Neumann series expansion,

$$(\mathbf{I} + A)^{-1} = \mathbf{I} - A + A^2 - \cdots \approx \mathbf{I} - A, \tag{28}$$

which holds for matrices with sufficiently small spectral norm, we obtain

$$(\mathbf{I} + \mathbf{y}\mathbf{y}^\top)^{-1} \approx \mathbf{I} - \mathbf{y}\mathbf{y}^\top, \qquad (\mathbf{I} + \mathbf{y}\tilde{\mathbf{y}}^\top)^{-T} \approx \mathbf{I} - \tilde{\mathbf{y}}\mathbf{y}^\top. \tag{29}$$

Substituting the above into the variational derivative yields

$$\frac{\delta\mathcal{R}}{\delta\mathbf{x}} \approx \mathbf{W}^*\big((\mathbf{I} - \mathbf{y}\mathbf{y}^\top)\mathbf{y}\big) - \mu\mathbf{W}^*\big((\mathbf{I} - \tilde{\mathbf{y}}\mathbf{y}^\top)\tilde{\mathbf{y}}\big), \tag{30}$$

and expanding the matrix products gives

$$\frac{\delta\mathcal{R}}{\delta\mathbf{x}} \approx \mathbf{W}^*\big(\mathbf{y} - \mathbf{y}(\mathbf{y}^\top\mathbf{y})\big) - \mu\mathbf{W}^*\big(\tilde{\mathbf{y}} - \tilde{\mathbf{y}}(\mathbf{y}^\top\tilde{\mathbf{y}})\big). \tag{31}$$

This derivation is rigorous and relies solely on the first-order Neumann expansion and standard matrix algebra, and it serves as the reference gradient when analyzing the approximation error of the attention-based form below.

To obtain a more intuitive subspace-membership interpretation, the inner-product terms $\mathbf{y}^\top\mathbf{y}$ and $\mathbf{y}^\top\tilde{\mathbf{y}}$ in equation 31 are heuristically replaced by softmax-normalized weights:

$$\frac{\delta\mathcal{R}}{\delta\mathbf{x}} \approx \mathbf{W}^*\Big(\mathbf{y} - \gamma\mathbf{y} \cdot \mathrm{softmax}(\mathbf{y}^\top\mathbf{y})\Big) - \mu\mathbf{W}^*\Big(\tilde{\mathbf{y}} - \gamma\tilde{\mathbf{y}} \cdot \mathrm{softmax}(\mathbf{y}^\top\tilde{\mathbf{y}})\Big), \tag{32}$$

where $\gamma$ is a scaling factor. Although this softmax replacement is originally motivated by an intuitive subspace-membership view, we next provide a closed-form error expression and an explicit sufficient condition under which the approximation from the Neumann-expanded gradient equation 31 to the attention-like form equation 32 is accurate.

**Approximation error analysis** We denote $V := \tilde{\mathbf{y}} \in \mathbb{R}^{d\times n}$, $C = [c_1, \ldots, c_n] := \mathbf{y}^\top\tilde{\mathbf{y}} \in \mathbb{R}^{n\times n}$, and $A = [a_1, \ldots, a_n] := \gamma\,\mathrm{softmax}(\mathbf{y}^\top\tilde{\mathbf{y}}) \in \mathbb{R}^{n\times n}$. We now analyze, for the optimal choice of $\gamma$, the approximation error incurred when Eq. equation 32 replaces $\tilde{\mathbf{y}}(\mathbf{y}^\top\tilde{\mathbf{y}})$ in Eq. equation 31 by $\gamma\tilde{\mathbf{y}} \cdot \mathrm{softmax}(\mathbf{y}^\top\tilde{\mathbf{y}})$. The same analysis applies to the approximation error incurred when $\mathbf{y}(\mathbf{y}^\top\mathbf{y})$ is replaced by $\gamma\mathbf{y} \cdot \mathrm{softmax}(\mathbf{y}^\top\mathbf{y})$. We further relax the scalar $\gamma$ to a diagonal matrix $\Gamma = \mathrm{diag}(\gamma_1, \ldots, \gamma_n)$. The corresponding approximation–error minimization problem is then defined as

$$\min_{\Gamma=\mathrm{diag}(\gamma_1,\ldots,\gamma_n)} \big\|VC - VA\Gamma\big\|_F^2. \tag{33}$$

Because the Frobenius norm is a sum of column-wise Euclidean norms, problem equation 33 decouples into $n$ independent one-dimensional problems:

$$\min_{\gamma_j\in\mathbb{R}} \big\|V(c_j - \gamma_j a_j)\big\|_2^2, \qquad j = 1, \ldots, n. \tag{34}$$

Thus it suffices to analyze the approximation error for a single column and then sum over $j$.

**Error bound.** Let $V \in \mathbb{R}^{d\times n}$ be fixed, and let $a, c \in \mathbb{R}^n$ denote a pair of weight vectors. Define

$$G := V^\top V \succeq 0, \qquad \langle x, y\rangle_G := x^\top G y, \qquad \|x\|_G := \sqrt{\langle x, x\rangle_G}, \tag{35}$$

and the $G$–cosine

$$\cos_G(a, c) := \frac{\langle a, c\rangle_G}{\|a\|_G\|c\|_G}, \qquad \text{for } \|a\|_G > 0,\ \|c\|_G > 0. \tag{36}$$

Consider the one-dimensional convex quadratic

$$f(\gamma) := \|V(c - \gamma a)\|_2^2 = (c - \gamma a)^\top G(c - \gamma a). \tag{37}$$

Writing $G = H^\top H$ with a symmetric square root $H := G^{1/2}$ (defined on the closure of $\mathrm{ran}(G)$), we have

$$f(\gamma) = \|Hc - \gamma Ha\|_2^2 = \|Hc\|_2^2 - 2\gamma\langle Ha, Hc\rangle + \gamma^2 \|Ha\|_2^2. \tag{38}$$

The minimizer and minimum are therefore

$$\gamma^* = \frac{\langle Ha, Hc\rangle}{\|Ha\|_2^2} = \frac{\langle a, c\rangle_G}{\|a\|_G^2}, \tag{39}$$

and

$$\min_{\gamma \in \mathbb{R}} f(\gamma) = \|Hc\|_2^2 - \frac{\langle Ha, Hc\rangle^2}{\|Ha\|_2^2} = \|c\|_G^2\Big(1 - \frac{\langle a, c\rangle_G^2}{\|a\|_G^2\|c\|_G^2}\Big) = \|Vc\|_2^2\big(1 - \cos_G^2(a, c)\big). \tag{40}$$

We thus obtain the following closed-form error identity:

$$\min_{\gamma \in \mathbb{R}} \big\|V(c - \gamma a)\big\|_2^2 = \|Vc\|_2^2\big(1 - \cos_G^2(a, c)\big), \qquad \gamma^* = \frac{\langle a, c\rangle_G}{\|a\|_G^2}. \tag{41}$$

In particular, if $\cos_G^2(a, c) \geq 1 - \varepsilon$ for some $\varepsilon \in [0, 1)$, then

$$\min_{\gamma \in \mathbb{R}} \big\|V(c - \gamma a)\big\|_2^2 \leq \varepsilon \|Vc\|_2^2. \tag{42}$$

Moreover, the minimum in equation 41 is zero if and only if $c \in \mathrm{span}\{a\} \oplus \ker(G)$, i.e., there exist $\lambda \in \mathbb{R}$ and $z \in \ker(G)$ such that $c = \lambda a + z$; when $G \succ 0$, this reduces to strict collinearity $c = \lambda a$.

Equations equation 41–equation 42 provide a closed-form error expression for replacing the linear weights $c$ by attention weights $a$ (plus optimal scaling). Summing over columns $j$ gives the corresponding Frobenius-norm error for problem equation 33, and this identity will be used to justify the gradient-to-attention mapping in Eq. equation 3 and its operator realization in Eq. equation 4.

**A sufficient condition for small approximation error.** We now derive an explicit sufficient condition under which the factor $\cos_G(a, c)$ in equation 41 is close to 1, so that the error bound equation 42 is very small.

Let $c \in \mathbb{R}^n$ be a column-wise linear weight vector. Using the translation invariance of softmax, we define a nonnegative shifted version

$$\hat{c} := c - \min_i c_i \mathbf{1} \geq 0, \qquad s := \mathbf{1}^\top \hat{c} > 0, \qquad \pi := \hat{c}/s \in \Delta_{n-1}, \tag{43}$$

and the corresponding attention weight

$$a := \mathrm{softmax}(\beta\hat{c}) \in \Delta_{n-1}, \tag{44}$$

where $\beta > 0$ is the temperature. Note that $\pi$ and $\hat{c}$ are strictly collinear:

$$\langle \pi, \hat{c}\rangle_G = \frac{\|\hat{c}\|_G^2}{s}, \qquad \|\pi\|_G = \frac{\|\hat{c}\|_G}{s}, \qquad \cos_G(\pi, \hat{c}) = 1. \tag{45}$$

Denote the index of the main peak by

$$i^\star = \arg\max_i \hat{c}_i, \tag{46}$$

the main-peak ratio by

$$\rho := \frac{\max_i \hat{c}_i}{s} \in \big[1/n, 1\big], \tag{47}$$

and the logit gap by

$$\Delta := \min_{i \neq i^\star}\big(\hat{c}_{i^\star} - \hat{c}_i\big) \geq 0. \tag{48}$$

Writing $a = \pi + \delta$, we first bound $\cos_G(a, \hat{c})$ in terms of

$$\varepsilon := \frac{\|\delta\|_G}{\|\pi\|_G}. \tag{49}$$

Using the Cauchy–Schwarz and triangle inequalities, we obtain

$$\langle a, \hat{c}\rangle_G = \langle \pi, \hat{c}\rangle_G + \langle \delta, \hat{c}\rangle_G \;\geq\; \|\pi\|_G\|\hat{c}\|_G - \|\delta\|_G\|\hat{c}\|_G = (1-\varepsilon)\,\|\pi\|_G\|\hat{c}\|_G,$$
$$\|a\|_G \leq \|\pi\|_G + \|\delta\|_G = (1+\varepsilon)\,\|\pi\|_G, \tag{50}$$

hence

$$\cos_G(a, \hat{c}) = \frac{\langle a, \hat{c}\rangle_G}{\|a\|_G\|\hat{c}\|_G} \;\geq\; \frac{1-\varepsilon}{1+\varepsilon}, \qquad \cos_G^2(a, \hat{c}) \;\geq\; \left(\frac{1-\varepsilon}{1+\varepsilon}\right)^2. \tag{51}$$

We next bound $\varepsilon$ by relating the $G$–norm to the Euclidean norm on the two-dimensional subspace $S := \mathrm{span}\{\pi, \delta\}$. Let

$$\lambda_{\min} := \lambda_{\min}(G|_S), \qquad \lambda_{\max} := \lambda_{\max}(G|_S), \qquad \kappa_S := \frac{\lambda_{\max}}{\lambda_{\min}} \geq 1. \tag{52}$$

Then for any $z \in S$,

$$\sqrt{\lambda_{\min}}\|z\|_2 \;\leq\; \|z\|_G \;\leq\; \sqrt{\lambda_{\max}}\|z\|_2. \tag{53}$$

In particular,

$$\|\delta\|_G \leq \sqrt{\lambda_{\max}}\|\delta\|_2, \qquad \|\pi\|_G \geq \sqrt{\lambda_{\min}}\|\pi\|_2 = \sqrt{\lambda_{\min}}\,\frac{\|\hat{c}\|_2}{s}, \tag{54}$$

so that

$$\varepsilon = \frac{\|\delta\|_G}{\|\pi\|_G} \leq \kappa_S\,\frac{s}{\|\hat{c}\|_2}\,\|\delta\|_2. \tag{55}$$

Using $s = \|\hat{c}\|_1 \leq \sqrt{n}\,\|\hat{c}\|_2$ (Cauchy–Schwarz), we further obtain

$$\varepsilon \;\leq\; \kappa_S\sqrt{n}\,\|\delta\|_2. \tag{56}$$

It remains to bound $\|\delta\|_2 = \|a - \pi\|_2$ in terms of $\rho$ and $\Delta$. For the softmax weights, we have

$$a_{i^\star} = \frac{1}{1 + \sum_{i \neq i^\star}\exp\left[-\beta(\hat{c}_{i^\star} - \hat{c}_i)\right]} \;\geq\; \frac{1}{1 + (n-1)\mathrm{e}^{-\beta\Delta}}, \tag{57}$$

so that

$$1 - a_{i^\star} \leq (n-1)\,\mathrm{e}^{-\beta\Delta}. \tag{58}$$

Moreover,

$$\sum_{i \neq i^\star} a_i^2 \leq \sum_{i \neq i^\star} a_i = 1 - a_{i^\star}, \tag{59}$$

which implies

$$\|a - e_{i^\star}\|_2^2 = (1 - a_{i^\star})^2 + \sum_{i \neq i^\star} a_i^2 \leq 2(1 - a_{i^\star}) \leq 2(n-1)\,\mathrm{e}^{-\beta\Delta}. \tag{60}$$

Similarly, for $\pi$ we have

$$\|\pi - e_{i^\star}\|_2^2 = (1 - \rho)^2 + \sum_{i \neq i^\star} \pi_i^2 \leq 2(1 - \rho). \tag{61}$$

By the triangle inequality,

$$\|\delta\|_2 = \|a - \pi\|_2 \leq \|a - e_{i^\star}\|_2 + \|\pi - e_{i^\star}\|_2 \leq \sqrt{2(1-\rho)} + \sqrt{2(n-1)\,\mathrm{e}^{-\beta\Delta}}. \tag{62}$$

Absorbing numerical constants into a universal constant $C > 0$, we obtain the bound

$$\varepsilon \;\leq\; C\,\kappa_S\left(\sqrt{1-\rho} + \sqrt{n-1}\,\mathrm{e}^{-\beta\Delta/2}\right). \tag{63}$$

Combining equation 51 and equation 63, we see that when the right-hand side of equation 63 is much smaller than 1 (i.e., the linear weights are sharply peaked with large $\rho$, the logit gap $\Delta$ is sufficiently large under the given temperature $\beta$, and $G$ is not severely ill-conditioned on $S$), we have $\cos_G(a, \hat{c}) \approx 1$ and hence $\cos_G^2(a, \hat{c}) \approx 1$. If, in addition, $G$ is insensitive to constant shifts

(for instance, $G\mathbf{1} = 0$ or we apply a $G$–orthogonal projection to remove the mean component), then $\cos_G(a, c) = \cos_G(a, \hat{c})$ and equation 63 directly controls $\cos_G(a, c)$.

Substituting this into equation 41–equation 42 shows that, under equation 63, the approximation error between the softmax-based form equation 32 and the Neumann-expanded gradient equation 31 is negligible. In summary, the identity equation 41 together with the bound equation 42 provides a closed-form error bound for the gradient-to-attention mapping in Eq. equation 3, and the sufficient condition equation 63 characterizes when this error bound is very small. Since Eq. equation 4 is an operator-level instantiation of the same mapping (via SPSA/MPSA), the same error identity and condition equation 63 also justify the approximation steps in Eq. equation 4.

## D    LIMITATION

Despite achieving competitive results across restoration tasks and affording transparent prompt–attention dynamics, one aspect merits further investigation: *computational efficiency*. Specifically, unrolling the gradient flow and stacking Transformer blocks increase the backpropagation memory footprint and wall-clock training time; at inference, the near-quadratic complexity of attention with respect to sequence length (or spatial resolution) can exacerbate latency. We view this as an engineering trade-off rather than a fundamental limitation, and it does not compromise our core conclusion of a variationally anchored, interpretable prompt–attention coupling; nevertheless, further optimization is warranted for resource- and latency-constrained deployments.

A second limitation lies in our current treatment of multi-scale processing. While the single-scale WBPT is fully white-box, the multi-scale extension WBPT$^\dagger$ introduces a learned pyramid aggregator for cross-scale fusion. This component remains black-box, even though the underlying attention and gradient-flow dynamics are white-box. From a variational perspective, a natural way to obtain a fully differentiable white-box pyramid is to use a multigrid V-cycle as the mathematical backbone: restriction and prolongation operators implement variational down/up sampling, while the STV-derived attention acts as the inter-grid operator on multi-channel features at each scale. Developing such a multigrid-style white-box pyramid, with STV-consistent cross-scale regularization explicitly encoded in these operators, is an important topic of ongoing work.

A third limitation concerns the scope of our experimental evaluation. In the present version, we systematically train and tune WBPT on the classic three-degradation all-in-one benchmark (denoising, deraining, dehazing), but do not yet provide full training and hyper-parameter search on larger unified protocols such as five-degradation or mixed-degradation settings, or on real-world benchmarks summarized in recent surveys. Extending WBPT to these more diverse settings and conducting a comprehensive study of its behavior under diverse degradation combinations is left for future work.

## E    QUALITATIVE RESULTS

We present additional qualitative results under the single-task setting to further demonstrate the effectiveness of *prompt-block*. The presented examples correspond one-to-one with the three quantitative single-task tables in the main text (Tables 2, 3, 4), serving to complement and visually illustrate the trends reported in the main paper.

## F    ALGORITHM

This part provides PyTorch-style pseudocode for the WBPT. Alg. 1 outlines the overall training loop with $T$-stage learnable gradient updates; Alg. 2 details one iteration stage with the SwinIR backbone and prompt injection; Alg. 3 specifies the prompted window attention and feedforward modules. Unless otherwise noted, we use the following defaults in the pseudocode: *epochs* = 120, *stages* $T = 10$, `embed_dim` = 96, `num_heads` = $[6, 6, 6]$, `window` = 8, `prompt_len` = 5, Adam with ($\beta_1$=0.9, $\beta_2$=0.999), learning rate $1 \times 10^{-4}$, MSE reconstruction loss, and $128 \times 128$ random crops with rotation/flip augmentations. The pseudocode focuses on core computations; engineering aspects (I/O, multi-GPU, logging, checkpointing) are omitted for brevity. Complexity expressions report the dominant terms (attention and MLP). Notation: $x_{\text{noisy}}/x_{\text{clean}}$ denote inputs/targets, $z$ the

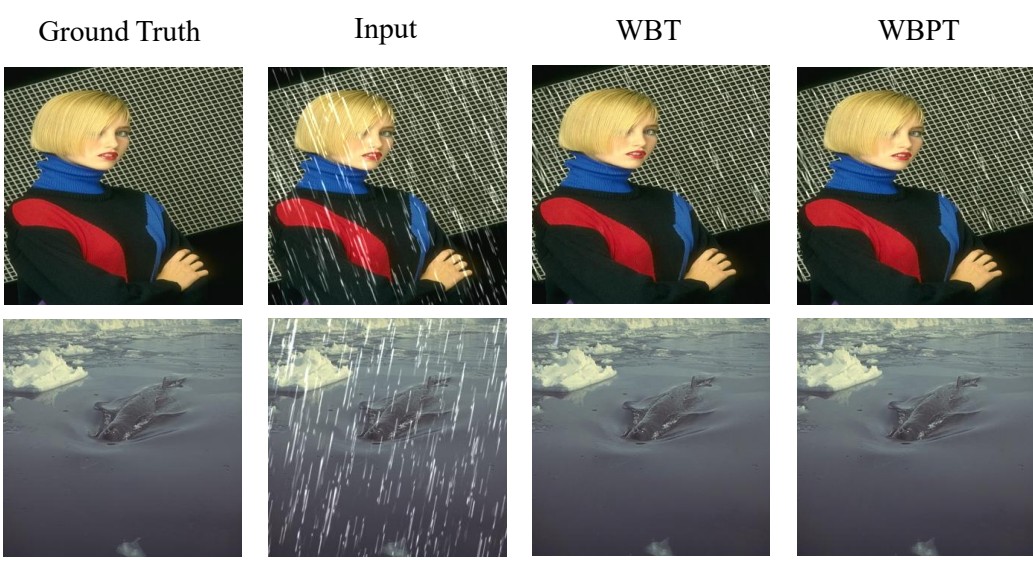

Figure 10: Deraining results for all-in-one methods.

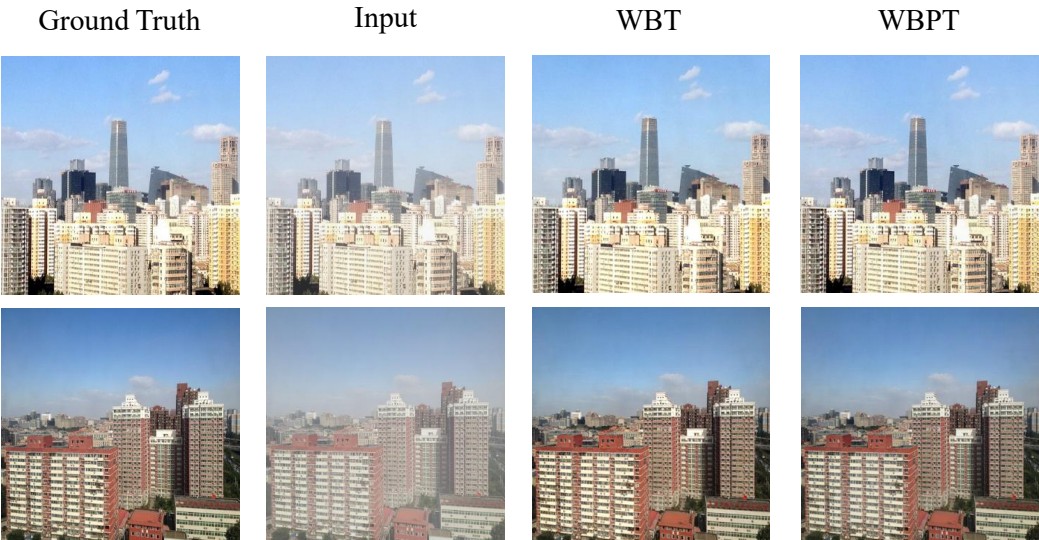

Figure 11: Dehazing results for all-in-one methods.

current state, $U_k$ the backbone output/prompt at stage $k$, and $t$ the iteration index. Prompt injection is fixed at block indices $\{2, 4, 6\}$ within the SwinIR backbone.

## G    TRANSFORMER AND PROMPT BLOCKS IN WBPT

As stated in Section 2.2 of the main manuscript, we present in Fig. 13 the block diagram of the Prompt block corresponding to $\xi_i$, and further elaborate on the implementation details of the Transformer block used within this Prompt block in Fig. 14. The Prompt block and the Transformer block follow the design and hyper-parameter settings outlined in Potlapalli et al. (2023) and Zamir et al. (2022), respectively.

| Ground Truth | Input | WBT | WBPT |
|---|---|---|---|

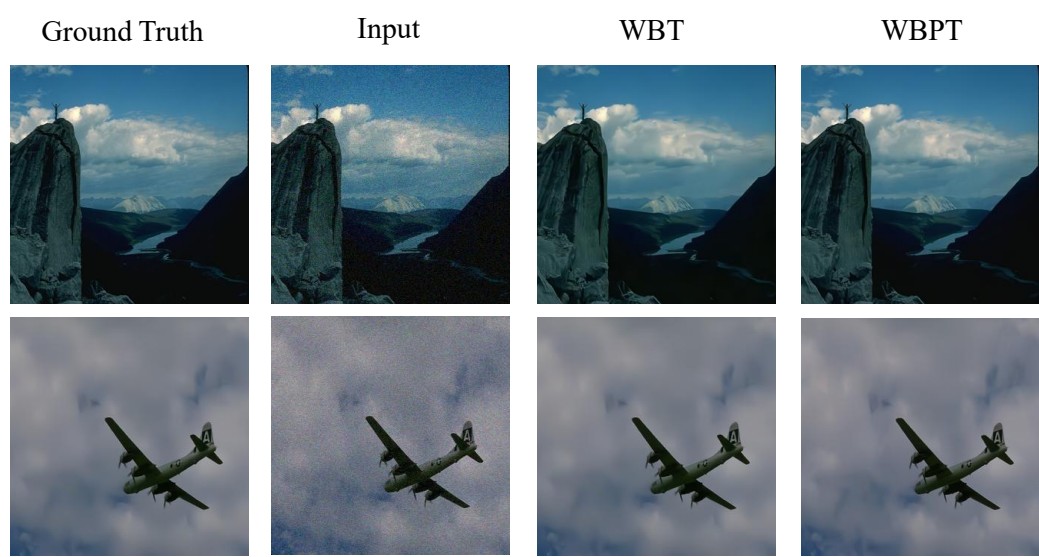

Figure 12: Denoising results for all-in-one methods.

---

**Algorithm 1:** PyTorch-style pseudocode for WBPT training (overall)

---

**Input:** Dataset $\mathcal{D}$; epochs $E$ (default 120); stages $K$ (default 10); hyperparameters and prompt config $P$

**Output:** Trained parameters $\theta^\star$; best validation metrics (PSNR/SSIM)

**Optimizer & Loss:** Adam($\beta_1$=0.9, $\beta_2$=0.999), lr $1 \times 10^{-4}$; reconstruction loss $\mathcal{L} = \text{MSE}$

**for** epoch $\leftarrow 1$ **to** $E$ **do**

    **foreach** $(\mathbf{x}_{\text{noisy}}, \mathbf{x}_{\text{clean}}) \in \text{DataLoader}(\mathcal{D})$ **do**

        $\mathbf{x}_0 \leftarrow \mathbf{x}_{\text{noisy}}$;  $\mathbf{W}_{\text{hist}} \leftarrow []$;  $\boldsymbol{\xi}_{\text{hist}} \leftarrow []$

        **for** $k \leftarrow 1$ **to** $K$ **do**

            $\{\mathbf{W}_i\}_{i=1}^N, \{\boldsymbol{\xi}_i\}_{i=1}^N \leftarrow \text{SwinIR}(\mathbf{x}_{k-1}; P, k)$    `# with optional prompt injection`

            Append $\{\mathbf{W}_i\}_{i=1}^N$ to $\mathbf{W}_{\text{hist}}$ and $\{\boldsymbol{\xi}_i\}_{i=1}^N$ to $\boldsymbol{\xi}_{\text{hist}}$; drop oldest if length $> 5$

            $\mathbf{x}_k \leftarrow \text{MPSA}(\mathbf{x}_{k-1}, \mathbf{y}=\mathbf{x}_{\text{clean}}, \{\mathbf{W}_i\}, \{\boldsymbol{\xi}_i\}, k)$

        $\mathcal{L} \leftarrow \text{MSE}(\mathbf{x}_K, \mathbf{x}_{\text{clean}})$

        optimizer.zero\_grad(); $\mathcal{L}$.backward(); optimizer.step()

    ValidateAndSaveIfBest($\theta$)    `# compute PSNR/SSIM on val, save best`

**Complexity:** $\mathcal{O}\big(E \cdot N \cdot K \cdot (F_{\text{swin}} + F_{\text{mpsa}})\big)$   `# Here N denotes the number of batches per epoch.`

---

## G.1 PROMPT BLOCK IN WBPT FRAMEWORK

### G.1.1 PROMPT BLOCK OVERVIEW

Given prompt components $\mathbf{P_c} \in \mathbb{R}^{N \times \hat{H} \times \hat{W} \times \hat{C}}$ and input features $\mathbf{F_l} \in \mathbb{R}^{\hat{H} \times \hat{W} \times \hat{C}}$, the prompt block refines the input via:

$$\hat{\mathbf{F}}_{\mathbf{l}} = \text{PIM}(\text{PGM}(\mathbf{P_c}, \mathbf{F_l}), \mathbf{F_l}) \tag{64}$$

The block contains two modules: the Prompt Generation Module (PGM) and the Prompt Interaction Module (PIM), detailed below.

---

**Algorithm 2:** One WBPT iteration stage (SwinIR with prompt injection)

---

**Input:** Current state $\mathbf{x}_{k-1}$; prompt config $P$; SwinIR depths $[2, 2, 2]$, channels $C=96$, window
  $M=8$
**Output:** $\{\mathbf{W}_i\}_{i=1}^N$, $\{\boldsymbol{\xi}_i\}_{i=1}^N$ (transformations and prompts for MPSA)
$\mathbf{f} \leftarrow \text{Conv3x3}(\mathbf{x}_{k-1})$                                                         # shallow feature
**for** $\ell \leftarrow 1$ **to** 3 **do**
    $(\mathbf{f}, \text{size}) \leftarrow \text{PatchEmbed}(\mathbf{f})$
    **for** $b \leftarrow 1$ **to** 2 **do**
        **if** $\text{ShouldUsePrompt}(\ell, b, P)$ **then**
            $\{\boldsymbol{\xi}_i\}_{i=1}^N \leftarrow \text{PromptGenBlock}(\mathbf{f}); \quad \mathbf{f} \leftarrow \mathbf{f} + \text{Inject}(\{\boldsymbol{\xi}_i\})$          # e.g., at
            fixed block indices $\{2, 4, 6\}$
        $\mathbf{f} \leftarrow \text{SwinTransformerBlock}(\mathbf{f}, \text{size})$
    **if** $\ell < 3$ **then**
        $\mathbf{f} \leftarrow \text{PatchMerging}(\mathbf{f}, \text{size})$
$\{\mathbf{W}_i\}_{i=1}^N$, $\{\boldsymbol{\xi}_i\}_{i=1}^N \leftarrow \text{ExtractTransformationsAndPrompts}(\mathbf{f})$
**return** $\{\mathbf{W}_i\}_{i=1}^N$, $\{\boldsymbol{\xi}_i\}_{i=1}^N$
**Complexity:** $\sum_{\ell=1}^3 \sum_{b=1}^2 \big(F_{\text{W-MSA}} + F_{\text{MLP}}\big) \approx \mathcal{O}(n_W \cdot B \cdot M^2 C + LC^2)$

---

---

**Algorithm 3:** Multi-Prompted Structure Attention (MPSA) with Learnable Data Consistency

---

**Input:** $\mathbf{x}_k \in \mathbb{R}^{B \times H \times W \times C}$; transforms $\{\mathbf{W}_i\}_{i=1}^N$; prompts $\{\boldsymbol{\xi}_i\}_{i=1}^N$; measurements/targets $\mathbf{y}$
**Output:** $\mathbf{x}_{k+1} \in \mathbb{R}^{B \times H \times W \times C}$
**Multi-Prompted Structure Attention:**
    **for** $i \leftarrow 1$ **to** $N$ **do**
        $\text{spsa}_i \leftarrow \text{SPSA}(\mathbf{x}_k \mid \mathbf{W}_i, \boldsymbol{\xi}_i, \gamma, \mu)$                                     # Eq. 4

    $\text{mpsa\_out} \leftarrow \begin{bmatrix} \mathbf{W}_1^* & \cdots & \mathbf{W}_N^* \end{bmatrix} \begin{bmatrix} \text{spsa}_1 \\ \vdots \\ \text{spsa}_N \end{bmatrix}$                  # Eq. 5

**Learnable Data Consistency:**
    $\Delta_\phi(\mathbf{x}_k, \mathbf{y}) \leftarrow \text{LearntGradient}(\mathbf{x}_k, \mathbf{y})$
**Gradient Flow Update:**
    $\mathbf{x}_{k+1} \leftarrow \big(\boldsymbol{I} - \eta \sum_{i=1}^N \mathbf{W}_i^* \mathbf{W}_i\big) \mathbf{x}_k - \eta\, \text{mpsa\_out} - \eta\, \Delta_\phi(\mathbf{x}_k, \mathbf{y})$
                                                                                            # Eq. 8
**return** $\mathbf{x}_{k+1}$
**Complexity:** MPSA $\mathcal{O}(N \cdot HWC^2)$; data consistency $\mathcal{O}(HWC^2)$

---

### G.1.2 PROMPT GENERATION MODULE (PGM)

PGM dynamically generates input-conditioned prompts. First, global average pooling (GAP) is applied on $\mathbf{F_l}$ to produce a channel-wise descriptor $\mathbf{v} \in \mathbb{R}^{\hat{C}}$. A $1 \times 1$ convolution and softmax yield prompt weights $w \in \mathbb{R}^N$:

$$w_i = \text{Softmax}(\text{Conv}_{1 \times 1}(\text{GAP}(\mathbf{F_l}))) \tag{65}$$

These weights modulate the learned prompt components to form the final prompt $\mathbf{P}$:

$$\mathbf{P} = \text{Conv}_{3 \times 3}\left(\sum_{c=1}^N w_i \mathbf{P}_c\right) \tag{66}$$

To support variable-resolution inputs, prompt components are upsampled to match the spatial size of $\mathbf{F_l}$ via bilinear interpolation.

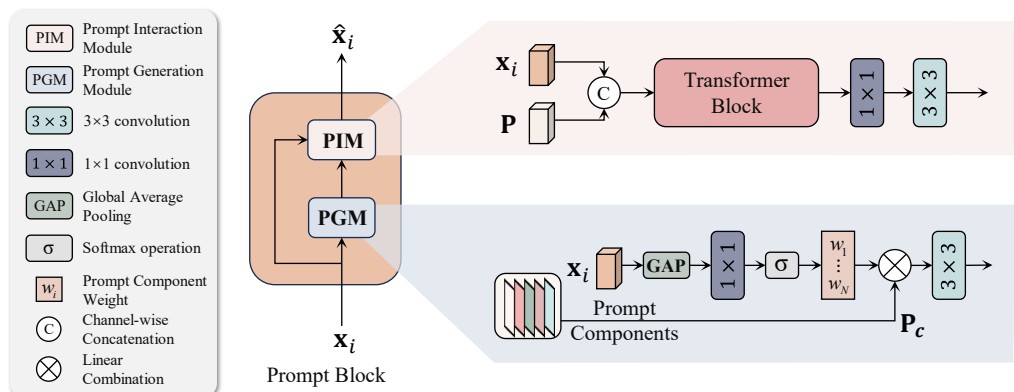

Figure 13: Overview of the Prompt block used in the WBPT framework. The Prompt block is composed of two sub modules,the Prompt Generation Module (PGM) and the Prompt Interaction Module (PIM).

### G.1.3    PROMPT INTERACTION MODULE (PIM)

PIM fuses the generated prompt $\mathbf{P}$ with features $\mathbf{F_1}$ via channel-wise concatenation, followed by a Transformer block:

$$\hat{\mathbf{F}}_1 = \mathrm{Conv}_{3\times3}(\mathrm{GDFN}(\mathrm{MDTA}([\mathbf{F_1}; \mathbf{P}]))) \tag{67}$$

### G.2    TRANSFORMER BLOCK IN WBPT FRAMEWORK

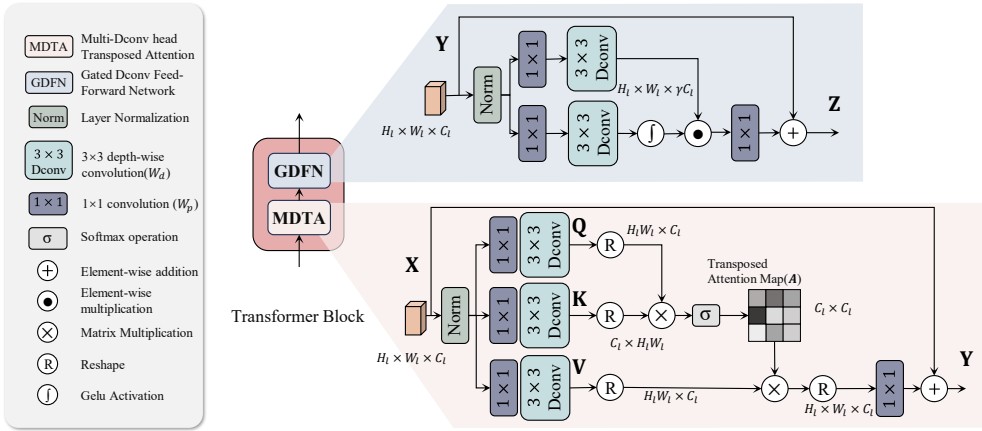

Figure 14: Overview of the Transformer block used in the Prompt block. The Transformer block is composed of two sub modules,the Multi Dconv head transposed attention module(MDTA) and the Gated Dconv feed-forward network(GDFN).

**MDTA Module.**    Let the input feature map be $\mathbf{X} \in \mathbb{R}^{H_l \times W_l \times C_l}$. MDTA first applies Layer Normalization, then projects the input to query ($\mathbf{Q}$), key ($\mathbf{K}$), and value ($\mathbf{V}$) tensors using a sequence of $1 \times 1$ pointwise convolution followed by $3 \times 3$ depth-wise convolution, all bias-free. To enable channel-wise attention, $\mathbf{Q}$ and $\mathbf{K}$ are reshaped to $\mathbb{R}^{H_l W_l \times C_l}$ and $\mathbb{R}^{C_l \times H_l W_l}$ respectively, resulting in a transposed attention map of shape $C_l \times C_l$ via dot-product interaction. Multi-head computation is performed in parallel.

**GDFN Module.** The GDFN submodule begins with a channel expansion using a $1 \times 1$ convolution by a factor $\gamma$. The expanded features are split into two parallel branches, each followed by a $3 \times 3$ depth-wise convolution. One branch passes through a GeLU activation while the other remains linear. The outputs are combined via element-wise multiplication, and finally projected back to the original channel dimension through a $1 \times 1$ convolution. Residual connections are maintained throughout the block.

**MDTA** performs self-attention along channels:

$$\mathbf{Y} = W_p \mathbf{V} \cdot \texttt{Softmax}(\mathbf{K} \cdot \mathbf{Q}/\alpha) + \mathbf{X}$$

**GDFN** transforms the result as:

$$\mathbf{Z} = W_p^0 \left( \phi(W_d^1 W_p^1 (\texttt{LN}(\mathbf{Y}))) \odot W_d^2 W_p^2 (\texttt{LN}(\mathbf{Y})) \right) + \mathbf{Y}$$

Here, $\texttt{LN}$ is layer normalization, $\phi$ denotes GELU activation, and $\odot$ is element-wise multiplication.

## H    REPRODUCIBILITY STATEMENT

We provide all necessary details to support reproducibility. All experiments are conducted on publicly available datasets, and the model architectures, hyperparameters, training protocols, and evaluation metrics are specified in the paper. We will release our codebase, training scripts, and pretrained checkpoints on GitHub upon acceptance.

## I    THE USE OF LARGE LANGUAGE MODELS (LLMS)

We used large language models only for light editorial assistance during manuscript preparation (grammar and wording refinement, minor style/formatting suggestions). No LLMs were used for research ideation, dataset curation, modeling, experiment design, analysis, or drafting substantive sections.

