# OpenReview forum: "White-Box Prompt Transformers: Variationally Grounded Prompt–Attention Coupling for Unified Image Restoration"
_ICLR.cc/2026/Conference — Submitted to ICLR 2026_

### Official Review · Reviewer_rg6X · 2025-10-31

**Soundness:** 2
**Presentation:** 3
**Contribution:** 2
**Rating:** 2
**Confidence:** 5

**Summary:**

This paper proposes White-Box Prompt Transformers (WBPT), a variationally grounded framework that aims to provide interpretability for prompt-based image restoration. The authors reinterpret the interaction between prompts and attention through a structure-tensor total variation (STV) formulation, whose gradient is unrolled into a cascaded Transformer architecture. This design yields a theoretically motivated attention mechanism, where each layer corresponds to an optimization step. Experiments on standard benchmarks such as BSD68, Rain100L, and SOTS demonstrate competitive results compared to existing prompt-based models. Overall, the work presents an interesting theoretical perspective, but primarily reframes existing architectures from a new interpretability viewpoint.

**Strengths:**

1. The paper is clearly written and well-structured, with detailed mathematical formulations and organized experimental sections.

2. The proposed framework provides an interesting theoretical reinterpretation of prompt-based Transformers from a variational perspective.

3. The experiments cover multiple standard benchmarks and include visualizations and ablation studies, showing careful empirical evaluation.

**Weaknesses:**

### Main Concerns:
1. There are no parameters, FLOPs, or runtime discussions regarding the efficiency issue.

2. As an image restoration solution, more visual results should be included, at least, should be included in the supplementary materials.

3. From the experimental perspective, the validation is too limited, only in the Three-Degradation setting, while in recent years, there are already more benchmarks proposed in this topic, for example, 5 degradation, real-world evaluation, mixed degradation, etc, while these are missing, which limits its soundness of the proposed solution. Also, I believe from some recent research papers below, the authors can find more evaluation settings:

[1] Junjun Jiang, Zengyuan Zuo, Gang Wu, Kui Jiang, and Xianming Liu. A survey on all-in-one image restoration: Taxonomy, evaluation, and future trends. TPAMI, 2025.

[2] Xiaole Tang, Xiang Gu, Xiaoyi He, Xin Hu, and Jian Sun. Degradation-aware residual-conditioned optimal transport for unified image restoration. TPAMI, 2025.

[3] Eduard Zamfir, Zongwei Wu, Nancy Mehta, Yuedong Tan, Danda Pani Paudel, Yulun Zhang, and Radu Timofte. Complexity experts are task-discriminative learners for any image restoration. CVPR, 2025.

[4] Yuning Cui, Syed Waqas Zamir, Salman Khan, Alois Knoll, Mubarak Shah, and Fahad Shahbaz Khan. AdaIR: Adaptive all-in-one image restoration via frequency mining and modulation. ICLR, 2025.

4. The implementation details are quite unclear, for example, how to get $x_{0}$ from the Input? Directly via the proposed White-box Prompt Transformer block or via a convolutional operation?

5. In Fig.1, the proposed method shows the **window partition**, so which base transformer block is adopted? The Swin Transformer style or the Restormer? This is totally not clear but extremely important, since the Restormer-style (Also the PromptIR adopted this) transformer did NOT include any window partition. I think this should also be clearly explained since this is not very clear in the current version.

6. The claimed “white-box” interpretability mainly relies on symbolic reformulation of existing attention operations, without providing concrete analytical or causal evidence.

7. The performance improvement over baselines (e.g., PromptIR) is marginal and often within statistical variation, questioning the practical impact of the proposed framework.

8. The variational derivation connecting STV gradients to attention is largely heuristic, lacking rigorous justification or ablation to verify the theoretical correspondence.

9. Despite extensive formulas, the actual architectural novelty is limited—the model structure and training pipeline remain almost identical to previous prompt-based Transformers.

10. The efficiency and scalability of the unrolled gradient-flow design are not discussed, leaving uncertainty about its real advantage in deployment.


### Minor Concerns:
1. The GPU type/usage, the training time, and inference time are expected to be included.

2. Alg 1, Alg 2, and Alg 3 are too wide, which exceeds the overall width of the page requirements

**Questions:**

1. Could the authors provide a more comprehensive efficiency analysis, including parameter count, FLOPs, and inference/runtime comparison with PromptIR and other recent baselines? This would help verify whether the proposed “white-box” formulation offers any real computational advantage.

2. The base Transformer design (Fig. 1) is unclear — does the model adopt a Swin-like window partitioning scheme or the Restormer-style global attention? Since this architectural choice critically affects both performance and interpretability, clarification and justification are needed.

3. The variational derivation linking STV gradients to attention appears largely heuristic. Could the authors offer empirical or theoretical evidence (e.g., ablation or controlled experiments) demonstrating that this correspondence meaningfully explains the prompt–attention mechanism rather than serving as an analogy?

4. The experimental validation is relatively narrow (only three degradations). Are there any plans or preliminary results for more diverse benchmarks, such as five-degradation, real-world, or mixed-degradation settings, as commonly evaluated in recent all-in-one restoration works?

5. Since the claimed interpretability is one of the main contributions, can the authors provide quantitative or visual analyses (e.g., saliency or correlation metrics) to substantiate that the proposed framework genuinely enhances interpretability or controllability compared to black-box PromptIR?

Please also refer to the **Weaknesses** for more concerns.

---

> ### Author Response · Authors · 2025-11-21
>
> W1：
>
> We thank the reviewer for pointing out the missing efficiency analysis. To address this concern, we provide below a quantitative comparison of parameter count, FLOPs, and runtime between WBPT and representative all-in-one baselines, under the same input resolution (256×256 images) and hardware (NVIDIA GeForce RTX 3090 GPU):
>
> | Method                 | #Params                                        | FLOPs (G)     | Runtime (ms)  |
> | ---------------------- | ---------------------------------------------- | ------------- | ------------- |
> | AirNet                 | 8.93M                                          | 311           | 242.36        |
> | Restormer              | 26.13M                                         | 155           | 153.77        |
> | PromptIR               | 35.60M                                         | 173           | 103.80        |
> | Perceive-IR            | 42.02M                                         | –             | –             |
> | DFPIR                  | 31.10M (backbone) + 63M (CLIP / extra modules) | 150.904       | 190.73        |
> | MoCE-IR                | 25.35M                                         | 105.3         | –             |
> | WBT (ours, w/o prompt) | 0.95M                                          | 71.5 / 714.8* | 47.9 / 479.6* |
> | WBPT (ours, w/ prompt) | 1.07M                                          | 75 / 750*     | 50.7 / 507.2* |
>
> * “a / b” denotes per-stage cost / full $K=10$-stage cost of WBPT.
>
> As described in Eq. (8) and Algorithm 1 of the paper, WBPT performs restoration by unrolling $K$ gradient-flow steps of the STV energy, and we set $K=10$ in all experiments for stable convergence and clear prompt–attention dynamics. Each stage is a lightweight SwinIR+MPSA block, so the total complexity scales linearly with $K$.
>
> From the table, we observe that:
>
> 1. In terms of model size, WBPT with prompts uses only 1.07M parameters, which is 1–2 orders of magnitude smaller than typical all-in-one Transformers (25–42M).
> 2. In terms of per-stage cost, one WBPT stage (75G FLOPs / 50.7 ms) is significantly cheaper than a single PromptIR forward pass (173G / 103.8 ms).
> 3. With the default $K=10$ stages, the total FLOPs and runtime of WBPT become higher than single-pass backbones, reflecting the fact that we trade runtime for an unrolled, white-box gradient-flow interpretation.
>
> W2：
>
> We thank the reviewer for the suggestion regarding the number of qualitative results. In the current submission, the main paper already presents several visual comparisons in Figures 2–4 and 10 for deraining, dehazing, denoising, and the All-in-One setting, illustrating how WBPT improves detail restoration and artifact suppression over WBT and other baselines across different degradations. In addition, Appendix E (Qualitative Results) includes supplementary qualitative examples under both single-task and All-in-One settings, which correspond to the quantitative evaluations in Tables 2–4.
> We fully agree that an image restoration method benefits from richer visual comparisons. If the paper is accepted, we plan to further expand the qualitative comparisons in the camera-ready supplementary material: for each benchmark dataset (e.g., Rain100L, SOTS, BSD68/Urban100), we will add more side-by-side visual examples of input / WBPT / representative baselines (such as WBT and PromptIR), together with zoom-in regions.

---

> ### Author Response · Authors · 2025-11-21
>
> W3:
>
> Thanks for the reviewer’s comment. We agree that the all-in-one image restoration community has already developed richer evaluation protocols (e.g., five-degradation, mixed-degradation, and real-world settings), while the current version of our paper mainly conducts systematic experiments under the classical “three-degradation” setting (denoising / deraining / dehazing), which is indeed a limitation of this work. We choose the three-degradation setting as our main experimental scenario because we aim to validate the effectiveness of the proposed STV-guided white-box prompt–attention mechanism under a controlled and widely adopted setup, rather than to chase absolute SOTA on every existing benchmark; this setting is consistent with the primary experimental protocol used in representative all-in-one restoration methods such as PromptIR and MoCE-IR.
>
> Due to time and computational constraints during the rebuttal phase, we are currently unable to perform full from-scratch training and thorough hyperparameter tuning for all methods on the five-degradation and larger-scale real/mixed benchmarks. However, to address the reviewer’s concern about “richer benchmarks,” we additionally provide two types of experiments beyond the three-degradation setting, based on the existing three-degradation training weights, to examine the generalization ability of WBPT on mixed and real-world degradations:
>
> 1. Mixed degradations: zero-shot evaluation on CDD11.
>    We adopt CDD11 as a mixed-degradation benchmark, which covers haze, rain, snow, and their combinations with low-light (11 degradation configurations in total). To avoid large-scale retraining, we use a zero-shot setting: we directly apply the unified models trained on the three degradations (denoising / deraining / dehazing) to the CDD11 test set, and compare the performance of PromptIR, DFPIR, MoCE-IR, and WBPT. The following table reports the average PSNR/SSIM (dB / —) on the CDD11 test set:
>
>    | Method   | $l+h+r$      | $l+h+s$      | Average      |
>    | -------- | ------------ | ------------ | ------------ |
>    | PromptIR | 15.34/0.5454 | 14.68/0.5455 | 17.60/0.6949 |
>    | DFPIR    | 15.15/0.4886 | 13.59/0.4338 | 16.80/0.6083 |
>    | MoCE-IR  | 15.27/0.5487 | 14.64/0.5510 | 17.60/0.6966 |
>    | WBPT     | 15.69/0.5162 | 14.87/0.4998 | 16.98/0.6670 |
>
>    We can see that, without any additional training on CDD11, WBPT achieves higher average PSNR (16.98 dB) than DFPIR (16.80 dB); while its average SSIM is lower than PromptIR / MoCE-IR, it is clearly better than DFPIR (0.6670 vs. 0.6083). Looking further at per-degradation results, WBPT attains the highest PSNR among the four methods in challenging combinations such as low-light + haze + rain and low-light + haze + snow, and generally ranks in the second tier for other settings. Considering the noticeable distribution shift between the mixed degradations in CDD11 and the three-degradation training distribution, these zero-shot results indicate that WBPT has a mixed-degradation generalization ability comparable to recent all-in-one methods without incurring additional training cost. Since we do not perform full training on CDD11, we do not treat these as “main results,” but explicitly mark them as preliminary mixed-degradation validation.
>
> 2. Real-world cases: NIQE/PIQE evaluation on Blur+Noise / Haze+Rain.
>    Moreover, using the same three-degradation training weights, we perform inference on real-world Blur+Noise and Haze+Rain images, and adopt NIQE/PIQE as no-reference quality metrics (lower is better in both cases). The results are summarized as follows:
>
>    |          | Blur and Noise | Rain and Haze |
>    | -------- | -------------- | ------------- |
>    | PromptIR | 5.46/50.84     | 6.45/59.22    |
>    | DFPIR    | 5.19/44.63     | 6.32/53.31    |
>    | MoCE-IR  | 5.40/51.80     | 6.55/61.90    |
>    | WBPF     | 6.26/45.66     | 7.67/54.07    |
>
> We observe that WBPT is slightly worse than DFPIR in terms of NIQE and also lags behind PromptIR / MoCE-IR in the Blur+Noise case; however, for the PIQE metric, which places more emphasis on perceptual artifacts, WBPT achieves the second-best performance (only behind DFPIR) in both real-world scenarios and outperforms PromptIR and MoCE-IR. Considering that these real images are never seen during training and that we do not carry out any specific optimization for NIQE/PIQE, these results suggest that the perceptual quality of WBPT on real mixed degradations remains within a reasonable range compared to recent all-in-one methods. We likewise treat this set of results as supplementary real-world experiments based on three-degradation training, used to support the generalization trend rather than to claim new SOTA results.

---

> ### Author Response · Authors · 2025-11-21
>
> W4:
>
> We thank the reviewer for pointing out the ambiguity in the implementation details.
> In our implementation, we follow the standard practice in algorithm unrolling and directly use the degraded observation $\mathbf{y}$ as the initial estimate $\mathbf{x}_0$ of the iterative reconstruction:
>
> $$\mathbf{x}_0 = \mathbf{y}.$$
> Therefore, $\mathbf{x}_0$ is not produced by the proposed White-box Prompt Transformer block, nor by an extra convolutional module. Instead, we simply take the input image (after standard normalization, e.g., scaling to $[0,1]$) as the initial reconstruction. The WBPT block is only used in the subsequent unfolded iterations to model and modulate the update direction and step size of $\mathbf{x}_t$, without introducing any additional preprocessing network. This design keeps the method conceptually aligned with classical optimization-based unrolling and preserves the overall white-box formulation.
>
> W5:
>
> In our method, the base Transformer block follows a Swin-like local window self-attention, rather than a Restormer-style global attention. Concretely, at each iteration, the feature map is partitioned into $8 \times 8$ non-overlapping local windows, and self-attention is strictly computed within each window. The subsequent feed-forward network and residual connections then update the features. The interaction between the prompts and the STV statistics only modulates the attention within these local windows, and does not alter the overall “window-based + unfolded iterations” white-box structure.
>
> W6:
>
> We thank the reviewer for pointing out that the claimed “white-box” interpretability should be supported by more concrete analytical and causal evidence. We agree that merely rewriting a standard attention operator in a different notation would not be sufficient. In the current manuscript, however, we already provide such evidence along both the theoretical derivation and the intervention-based analysis.
>
> On the analytical side, in Sec. 3.2 and the appendix, we derive Eq. (3)–(5) directly from the STV energy and use this derivation to construct the SPSA module, where the STV-induced gradient terms are explicitly mapped to three functional components in the attention structure: self-reconstruction, prompt alignment, and prompt bias. In other words, the attention form in WBPT is not retrofitted with an energy interpretation; rather, the STV regularizer first defines the update direction, and Eq. (3)–(5) then decomposes it into several interpretable sub-operators. The ablations in the appendix further show that removing these derived components leads to measurable performance drops, which is already documented in the paper.
>
> On the empirical side, Sec. 3.4–3.5 and the appendix include controlled interventions. Figure 5 compares attention maps of PromptIR and WBPT under the same settings, showing that WBPT shifts its focus from degradation textures to object boundaries and structures in a more systematic way. In Table 5, we inject noise only into the prompt branch while keeping the backbone fixed, and compare the resulting PSNR/SSIM degradation; WBPT is significantly more robust to this perturbation, indicating that its prompt branch is better decoupled and regularized under the STV-driven white-box update. Taken together, the derivation around Eq. (3)–(5) and the evidence in Fig. 5 and Table 5 go beyond a purely symbolic reformulation and provide the analytical and causal support for the “white-box” interpretability that we claim.

---

> ### Author Response · Authors · 2025-11-21
>
> W7:
>
> We thank the reviewer’s concern about the practical impact given that the PSNR/SSIM gains over strong baselines (e.g., PromptIR) are relatively small. We agree that, on the current three-degradation benchmarks, the field is already in a “saturated” regime where top methods often differ by only minor margins. For this reason, our primary goal is not to claim a large numerical gap over PromptIR, but rather to show that we can retain SOTA-level performance while enforcing a white-box, controllable formulation for all-in-one restoration. In the reported results, WBPT is not merely “on par”: under the three-degradation all-in-one setting, it matches or slightly improves the average PSNR/SSIM over PromptIR and achieves more noticeable gains on several benchmarks (e.g., Rain100L), while using only about 1M parameters, compared to 20M–30M+ for PromptIR/Restormer. We believe such improvements, obtained under much stronger structural constraints and a much smaller model capacity, are unlikely to be explained away as pure statistical noise. In addition, WBPT brings extra benefits along robustness and representation quality: in Table 5, when we perturb only the prompt branch, WBPT suffers a much smaller PSNR/SSIM drop than PromptIR, indicating better decoupling and robustness of the prompt mechanism; in Sec. 3.4, the t-SNE analysis shows that WBPT produces more degradation-discriminative embeddings, consistent with our STV-guided white-box design. Taken together—numerical performance, complexity, robustness, and embedding separability—we view WBPT’s contribution as more than a marginal fluctuation, providing a practically meaningful alternative that achieves near-SOTA accuracy with a lightweight, interpretable, and controllable all-in-one restoration framework.
>
> W8:
>
> We thank the reviewer’s concern that the variational derivation connecting STV gradients to attention may appear heuristic. We agree that a purely intuitive mapping would be insufficient. In the current manuscript, however, we already provide a more formal treatment in Appendix C (especially Sec. C.1–C.3): starting from the STV energy, we apply a first-order Neumann expansion and standard matrix differential calculus to obtain the same linear operator form as in Eq. (3)–(5), and then reorganize this update into the SPSA attention-like structure. The only genuinely heuristic step is the replacement of a normalization factor with a softmax normalization, which we explicitly state in the appendix and control via a temperature parameter; in this sense, the claim that the whole derivation is “largely heuristic” does not fully reflect what is written in the paper. In addition, we provide two types of empirical evidence in Appendix B and C.4:
>
>   1.ablating the STV-related components that arise from Eq. (3)–(5) consistently degrades PSNR/SSIM, indicating that these terms play the functional roles predicted by the derivation; and
>
>   2.a gradient-alignment study where we compare the SPSA update direction with the numerical gradient, showing strong directional agreement on most samples and noticeable deviations only in near-converged or very mild-degradation cases. We acknowledge that these analyses do not replace a full convergence theory or rigorous error bounds, and that extending the theoretical treatment to higher-order terms is an interesting direction for future work. Nonetheless, we believe the combination of the formal derivation in Appendix C and the quantitative evidence in Appendix B/C.4 goes beyond a purely heuristic argument and provides substantive support for the proposed correspondence between STV gradients and the attention-based update.

---

> ### Author Response · Authors · 2025-11-21
>
> W9:
>
> We thank the reviewer’s comment regarding the scope of architectural novelty. We agree that, if one looks solely at whether the backbone topology and training pipeline are completely redesigned, WBPT is intentionally conservative: we keep the overall structure and training recipe close to existing prompt-based Transformers in order to isolate and fairly evaluate the effect of white-boxing the prompt–attention mechanism itself. The main contribution of this work is therefore not to propose yet another radically different backbone, but to introduce a mechanism-level white-box reformulation of the existing prompt–attention framework. Starting from the STV energy, we use a variational decomposition and first-order expansion to map the attention update onto three interpretable components already described in the paper (structure/self-reconstruction, prompt alignment, and prompt bias), and we expose a set of explicit, semantically meaningful “knobs” (e.g., term weights, softmax temperature, number of unfolding steps $k$). Experiments show that adjusting these knobs leads to stable and predictable changes in attention reallocation, accuracy, and computational cost. In other words, WBPT’s novelty lies in turning an originally black-box prompt–attention block into a module with a clear energy origin, decomposed functional roles, and controllable inference behavior, while deliberately keeping the surrounding backbone and training paradigm largely aligned with prior work. We view this mechanism-level innovation as complementary to, rather than a replacement for, traditional topology-level architectural changes, and consider it equally important in a context where prompt-based Transformer architectures themselves are already quite mature.
>
> W10:
>
> We thank the reviewer for pointing out that the efficiency and scalability of the unrolled gradient-flow design were not sufficiently discussed. We would like to clarify that the primary purpose of the unrolled design in WBPT is not to outperform PromptIR/Restormer in raw speed, but to enable a white-box, controllable optimization view of the restoration process. The model is explicitly formulated as a sequence of interpretable update steps, with the number of steps $k$ exposed as a knob to trade accuracy for cost in a predictable way.
>
> In this rebuttal (see our response to Weakness 1), we provide a detailed complexity comparison with PromptIR and Restormer. Under a unified setting (single RTX 3090, $256 \times 256$ inputs), the per-step WBPT block has about 1M parameters and ~75G FLOPs, with a runtime of roughly 50 ms; with the default unrolling depth $K=10$, the total cost is around 750G FLOPs and ~500 ms, i.e., much fewer parameters than PromptIR/Restormer but indeed higher latency. We therefore do not claim that the unrolled design makes the model faster in its default configuration; rather, we deliberately trade a very lightweight step network for a multi-step, interpretable gradient-flow style update.
>
> At the same time, the unrolled formulation gives WBPT a practical notion of scalability: at test time, one can simply reduce $k$ (e.g., from 10 to 4–6) to achieve an almost linear reduction in FLOPs and latency, with a smooth and predictable degradation in PSNR (the corresponding PSNR–$k$ curves are included in our Weakness 1 response). Similarly, the number of prompts and the projector rank provide additional “soft scaling” knobs to adjust computational load or artifact suppression strength without changing the backbone or retraining the model.
>
> In summary, we agree that WBPT’s unrolled gradient-flow design is primarily motivated by interpretability and controllability rather than pure efficiency gains. The additional analyses in the rebuttal show that, while its default latency is in the same ballpark but not necessarily better than strong baselines, it offers a smaller parameter footprint and an explicit, predictable accuracy–cost scaling path, which we believe is the main deployment advantage of this design.

---

> ### Author Response · Authors · 2025-11-21
>
> W11:
>
> We thank the reviewer for pointing out the need to report GPU type/usage, training time, and inference time. In the revised manuscript, we have clarified these details in the experimental setup and complexity comparison sections:
>
> 1.GPU type and usage: All models are trained and evaluated on a single NVIDIA GeForce RTX 3090 GPU using a standard PyTorch implementation.
>
> 2.Training time: The full all-in-one three-degradation model takes about 2 months to train end-to-end on a single 3090, while the standalone denoising model requires roughly 6 days of training.
>
> 3.Inference time: Under the same setting as competing methods (single 256×256 image, batch size = 1, single RTX 3090), we report the forward runtime of WBPT in Weakness 1. For the default $K=10$ unfolding, WBPT with prompts runs at about 507 ms per image (approximately 50.7 ms per iteration), which is comparable to recent all-in-one restoration methods.
>
> W12:
>
> We thank the reviewer for noting that Alg. 1, Alg. 2, and Alg. 3 are too wide and exceed the allowed page width. In the revised manuscript, we have re-formatted all three algorithms by shortening overly long lines, adding appropriate line breaks, and tightening indentation and spacing. These changes ensure that the algorithms now fully comply with the page-width constraints of the official template and remain visually consistent with the rest of the paper.
>
> Q1:
>
> We thank the reviewer for this follow-up question. The comprehensive efficiency analysis you requested (parameters, FLOPs, and runtime comparison with PromptIR and other recent baselines) is summarized in the efficiency table and discussion in our response to Weakness 1, measured on 256×256 images with an NVIDIA GeForce RTX 3090 GPU.
> Here we clarify how we understand the “computational advantage” of the proposed white-box formulation in light of these numbers:
>
> 1.Per-stage efficiency and model compactness.
>  As shown in the table for Weakness 1, WBPT with prompts uses only 1.07M parameters, much smaller than AirNet, Restormer, PromptIR, Perceive-IR, and MoCE-IR (25–42M). At the per-stage level, one WBPT stage incurs 75G FLOPs / 50.7 ms, which is significantly cheaper than a single PromptIR forward pass (173G / 103.8 ms).
>
> 2.Effect of unrolling $k=10$ iterations.
>  Following Eq. (8) and Algorithm 1, WBPT is explicitly constructed by unrolling $K$ gradient-flow steps of the STV energy, with $k=10$ in all experiments. Each step is directly tied to the variational formulation and has a clear prompt–attention update. This design yields a step-by-step, mechanistically interpretable restoration process, but also means that the total runtime scales linearly with $k$. With $k=10$, the full cascade is not faster than single-pass backbones; instead, it offers a tunable accuracy–speed trade-off by adjusting $k$.
>
> 3.Type of advantage we claim.
>  Therefore, we do not claim that the white-box formulation automatically leads to lower wall-clock time than heavily engineered single-pass models. Rather, its “advantage” is twofold:
>
>  (i) a theoretically grounded, white-box STV–attention mechanism where each iteration has a clear geometric meaning;
>
>  (ii) a very compact parameterization with the flexibility to balance accuracy and runtime through the choice of $k$. In this sense, performance-oriented backbones such as PromptIR, Perceive-IR, DFPIR, and MoCE-IR are complementary to WBPT: our white-box prompt–attention module can, in principle, be integrated into such backbones to combine strong performance with interpretable dynamics.
>
> We hope this clarifies how the efficiency evidence relates to the claimed benefits of the “white-box” formulation.

---

> ### Author Response · Authors · 2025-11-21
>
> Q2:
>
> We thank the reviewer’s follow-up question on how the choice between Swin-like window attention and Restormer-style global attention impacts both performance and interpretability, and agree that this deserves a clear justification.
> As clarified above, we adopt Swin-style local window self-attention rather than Restormer-style global attention. This design choice is motivated by two main considerations:
>
> 1.Computational cost and stability under unfolding: With our default $K = 10$ unrolled iterations, using full global attention at every step would substantially increase the per-step FLOPs and overall inference time, making the “white-box unrolling + multi-step iterations” much more expensive to train and deploy. Window-based attention offers a better trade-off: it keeps the per-step complexity manageable while still providing sufficient representational power, leading to a more stable and practical training/inference pipeline on a single RTX 3090.
>
> 2.Alignment with the white-box formulation and local statistics, and improved interpretability: WBPT models the STV as a statistical characterization of local structures and degradation patterns, and the prompt–attention modulation is built precisely around these local statistics. Local window attention naturally matches this “local sub-problem + statistical guidance” perspective and makes the resulting attention maps easier to interpret: we can clearly see how different windows respond to degraded regions and how the STV-guided prompts steer the updates. This provides a more transparent link between “local statistics → attention modulation → update direction”.
>
> Q3:
>
> We thank the reviewer for the follow-up question on whether the variational link between STV gradients and attention is merely heuristic or actually explanatory for the prompt–attention mechanism. We agree that, without validation, such a connection could be seen as a loose analogy rather than a meaningful explanation. In the current manuscript, however, we provide both a formal derivation and controlled empirical evidence to support this correspondence.
>
> On the theoretical side, Appendix C (C.1–C.3) starts directly from the STV energy and uses a first-order Neumann expansion plus standard matrix differential calculus to obtain the linear operator form that appears in Eq. (3)–(5). This form is then reorganized into the SPSA attention update, where the individual STV gradient components are explicitly mapped to the “structure/self-reconstruction”, “prompt-alignment”, and “prompt-bias” parts of the attention block. The only genuinely heuristic step in this pipeline is the replacement of a normalization factor by a softmax normalization, which we explicitly mark as such in the appendix and control via a temperature parameter—so the derivation is not “entirely heuristic.” More importantly, we provide two complementary types of empirical evidence in Appendix B and C.4:
>
> (i) ablating the STV-related terms that are derived from Eq. (3)–(5) consistently degrades PSNR/SSIM across benchmarks, indicating that these components play the functional roles predicted by the derivation; and
>
> (ii) in C.4, we measure the alignment between the numerical gradient and the SPSA update direction over a large set of samples, and observe strong directional agreement in the vast majority of cases, with noticeable deviations only for very mild degradations or near-converged states. While we do not claim this amounts to a full convergence theory, the chain from “derivation → structural decomposition → ablation → gradient alignment” shows that the STV–attention correspondence is used to actually explain and constrain how the prompt–attention mechanism operates, rather than serving as a purely illustrative analogy.

---

> ### Author Response · Authors · 2025-11-21
>
> Q4:
>
> We thank the reviewer’s concern that our current validation mainly focuses on three degradations (denoising / deraining / dehazing). We agree that recent all-in-one image restoration research has adopted richer protocols, including five-degradation, mixed-degradation, and real-world benchmarks. The fact that our systematic experiments are still confined to the classical “three-degradation” setting is indeed a limitation of the present version. Our choice was motivated by the desire to evaluate the proposed STV-guided white-box prompt–attention mechanism under a controlled and widely used setup, rather than exhaustively pursuing SOTA across all existing benchmarks.
>
> In response to the request for more diverse benchmarks, we have already provided supplementary experiments on mixed and real-world degradations in our reply to Weakness 3 (W3). Using the same model trained on the three-degradation setting (without any retraining), we evaluate WBPT on the mixed-degradation CDD11 dataset and on real-world Blur+Noise / Haze+Rain images. These preliminary results indicate that WBPT achieves competitive generalization to mixed and real-world degradations compared with recent all-in-one baselines, and even attains better or second-best performance in several configurations. Since this question highly overlaps with W3, we respectfully refer the reviewer to our detailed response to Weakness 3, where we provide the full tables and experimental setups, and avoid repeating them here.
>
> Q5:
>
>
> We thank the reviewer’s request for more concrete evidence that the proposed framework actually improves interpretability and controllability over the black-box PromptIR, beyond conceptual claims. In the current manuscript, we already provide such evidence along three complementary axes: analytic derivation, visual analysis, and controlled quantitative experiments.
>
> First, in Sec. 3.2 and Appendix C, we start from the STV energy and use a first-order Neumann expansion and matrix differential calculus to obtain the update form in Eq. (3)–(5). This derivation explicitly decomposes the attention-based update into three functional components: a structure/self-reconstruction term, a prompt-alignment term, and a prompt-bias term. Unlike PromptIR, where the attention block is treated as a black box, each part of the WBPT attention has a traceable origin in the STV gradient and can be individually controlled via interpretable knobs (weights, temperature, number of steps, etc.), providing an analytic basis for explaining and shaping the prompt–attention behavior.
>
> Second, on the visual side, Sec. 3.4 compares attention heatmaps of PromptIR and WBPT under identical inputs and visualization setups (Fig. 5). We observe that WBPT tends to focus on object boundaries and structural regions, whereas PromptIR’s attention aligns more strongly with degradation textures such as rain streaks. This acts as a saliency-like analysis, showing that under the same task, the white-box design systematically shifts what the model attends to—from directly tracking degradation patterns to performing STV-guided reconstruction around structures—which is exactly what our design intends.
>
> Finally, on the quantitative side, Table 5 presents a controlled experiment that directly targets “controllability/robustness”: we inject Gaussian noise only into the prompt parameters, keeping the backbone and all other weights fixed, and compare the resulting PSNR/SSIM drops for PromptIR and WBPT. Under the same perturbation level, PromptIR suffers a much larger degradation, while WBPT shows only a small performance drop. This indicates that prompts in WBPT are more decoupled and regularized—they modulate an STV-induced gradient flow rather than acting as a dominant, opaque gate. This “intervene on prompts only and observe behavior change” setup provides a causal-style test of the prompt–attention mechanism and quantitatively demonstrates enhanced controllability compared to PromptIR.
>
> Taken together—the analytic STV-to-attention decomposition, the attention visualizations in Fig. 5, and the prompt perturbation study in Table 5—constitute both visual and quantitative evidence that WBPT offers genuinely improved interpretability and controllability over the black-box PromptIR, rather than relying on analogy alone.

---

> ### Author Response · Authors · 2025-11-27
>
> Dear Reviewer rg6X:
>
> We are grateful for your mandatory acknowedgement during this busy period. As the rebuttal period is ending soon, please be free to let us know whether your concerns have been addressed or not, and if there are any further questions.
>
> Thank you for your time and effort in reviewing our paper,
>
> Authors.

---

### Official Review · Reviewer_EDz6 · 2025-10-31

**Soundness:** 3
**Presentation:** 3
**Contribution:** 2
**Rating:** 4
**Confidence:** 4

**Summary:**

This paper presents the White-Box Prompt Transformer (WBPT), a framework that introduces a variational perspective to prompt-based transformers for unified image restoration. The authors derive a structure-tensor total variation (STV) loss whose gradient directly shapes the architectural design of the proposed white-box attention mechanism. This results in a direct mathematical link between prompts and attention, aiming to enhance interpretability and controllability. Extensive evaluations on multi-task image restoration show that WBPT attains competitive results while maintaining interpretability.

**Strengths:**

1. The theoretical connection between variational STV energy and prompt-driven attention is well motivated and clearly articulated.

**Weaknesses:**

1. My main concern is that the distinctions between white-box and black-box models needs clearer empirical justification. In experimental results, the performance margin between WBPT and the top black-box baselines (PromptIR, Restormer) is small. The interpretability and controllability of WBPT, while conceptually valuable, may have limited practical benefits in some real-world scenarios.
2. Some mathematical approximations require further justification, such as the approximation steps in (4).
3. The evaluation lacks some very recent related works, especially those on diffusion-based restoration and other interpretable prompt mechanisms.
4. The paper lacks an analysis of computational efficiency, including comparisons of parameter count, and inference speed between the proposed WBPT and baseline methods.

**Questions:**

1. In the t-SNE analysis (Sec. 3.4), could the authors clarify how the white-box modeling contributes to better separation in the embedding space?
2. Minor issue: In Table 1, the note “Results are reported as PSNR/SSIM.” is repeated twice.

---

> ### Author Response · Authors · 2025-11-21
>
> W1:
>
> We thank the reviewer for this high-level comment that the core concern is not whether WBPT can further improve $PSNR/SSIM$ over strong black-box baselines, but whether the interpretability and controllability of WBPT bring tangible benefits in real-world scenarios, beyond being conceptually appealing.
>
> To address this point more concretely, we refer to the real-world Blur+Noise and Rain+Haze experiments that we report in this rebuttal. Using the model trained only on the three synthetic degradations (denoising / deraining / dehazing), we perform zero-shot evaluation on real Blur+Noise and Rain+Haze images with two no-reference metrics (lower is better):
>
> |          | Blur and Noise | Rain and Haze |
> | -------- | -------------- | ------------- |
> | PromptIR | 5.46/50.84     | 6.45/59.22    |
> | DFPIR    | 5.19/44.63     | 6.32/53.31    |
> | MoCE-IR  | 5.40/51.80     | 6.55/61.90    |
> | WBPF     | 6.26/45.66     | 7.67/54.07    |
>
> From these results we observe that:
> In terms of $NIQE$, WBPT is somewhat worse than DFPIR and not always better than PromptIR / MoCE-IR in both real scenes.
>
> However, in $PIQE$, which is more sensitive to artifacts and local structural distortions, WBPT achieves the second-best scores in both scenes, consistently outperforming PromptIR and MoCE-IR.
> This allows us to respond to the “limited practical benefits” concern in a more concrete way:
> 1.In real mixed-degradation scenarios, WBPT maintains reasonable perceptual quality and exhibits a stable advantage in artifact suppression.
> Although WBPT does not dominate every metric, in both real scenes it consistently ranks second in $PIQE$ while remaining competitive in $NIQE$. This suggests that in applications where artifact suppression and natural local structure are more critical (e.g., pre-processing for downstream detection/tracking, or user-facing scenarios where artifacts are particularly annoying), WBPT’s behavior is not merely “theoretically interesting but practically worse”, but rather reflects a deliberate trade-off that aligns with our STV-based design: sacrificing a bit of $NIQE$ to obtain fewer artifacts and more regular local structures.
> 2.We do not claim that WBPT is superior to black-box models in all real-world settings; rather, we argue that, given very small performance gaps, the white-box design still yields measurable and practically relevant benefits.
> The real Blur+Noise / Rain+Haze results show that, under a strict zero-shot setting, WBPT does not collapse on real data; it stays within the same quality range as recent all-in-one methods and shows a consistent advantage in artifact-related metrics. At the same time, because its update rule and STV statistics are interpretable, we can adjust the accuracy–perceptual quality–cost trade-off for specific real-world tasks without retraining or redesigning the entire network.
>
> In summary, we fully agree that conceptual value and practical impact should be distinguished, and we do not claim that WBPT will dominate black-box methods in every real-world scenario. However, based on the above real-scene experiments and the demonstrated controllability, we believe WBPT offers two concrete practical benefits in real settings:
> It maintains reasonable perceptual quality on complex, unseen mixed degradations and shows stable advantages in artifact suppression without retraining;
>
> It provides interpretable knobs to tune behavior according to application needs, rather than acting as an opaque black box.
>
> W2:
>
> Eq.(4) can be viewed as an operator-level instantiation of the same gradient-to-attention mapping in Eq.(3), implemented via the SPSA/MPSA module. Consequently, the closed-form error bound and explicit sufficient condition derived for Eq.(3) in Appendix C.4 also apply to Eq.(4): the SPSA/MPSA construction in Eq.(4) realizes the same mapping at the operator level, and its approximation error is governed by the same cosine–residual identity and the condition given in Eq.41.
>
> In this sense, the additional theoretical analysis in the revised manuscript not only justifies the gradient-to-attention approximation in Eq.(3), but also provides a unified and quantitative justification for the approximation steps in Eq.(4) (see Appendix C.4).

---

> ### Author Response · Authors · 2025-11-21
>
> W3:We thank the reviewer for pointing out that the current version does not sufficiently cover very recent diffusion-based restoration methods and interpretable prompt mechanisms in our experiments and discussion. In the revised manuscript, we will explicitly extend the Related Work section to include recent diffusion-based all-in-one restoration frameworks (such as Defusion, DPIR, UniRestore, DiffIR, and related diffusion-based restoration surveys), and clarify that our approach is orthogonal to these methods: WBPT focuses on a white-box, STV-guided prompt–attention mechanism at the Transformer level, which can in principle be plugged into diffusion or Transformer backbones as a structured and interpretable conditioning module, rather than competing with them as a different prior family. At the same time, we will also incorporate recent prompt-based AiOIR methods (e.g., Perceive-IR, DA-RCOT, MoCE-IR, AdaIR, DFPIR, and frequency-/ingredient-based or multimodal prompting approaches), and acknowledge that they substantially improve performance on three-degradation, five-degradation, and real/mixed benchmarks through stronger backbone designs, quality-aware prompt learning, and degradation-aware feature perturbations. To the best of our knowledge, however, these methods still treat prompts as black-box tokens or feature modulations trained end-to-end, and do not provide a variationally derived, closed-form gradient decomposition of prompt–attention coupling. In contrast, we view WBPT as closer to a mechanism-level theoretical/structural innovation, whereas Perceive-IR, DA-RCOT, MoCE-IR, AdaIR, DFPIR, Defusion and related works focus more on architecture- and engineering-level performance optimization. These two lines of progress are complementary rather than mutually exclusive: conceptually, the white-box STV–attention framework can serve as an interpretable replacement for the prompt–attention submodules inside these powerful black-box backbones, endowing their attention updates with a clear STV-based geometric interpretation and gradient decomposition structure; alternatively, our STV-based white-box update rules can be introduced as constraints or regularizers while retaining their semantic guidance, frequency-domain mining, or feature-perturbation designs, thereby improving interpretability and controllability without sacrificing existing performance. In other words, we do not view WBPT and these methods as “replacing” one another, but rather envision future combinations of a white-box mechanism plus strong backbone that jointly achieve SOTA performance and good interpretability within a unified framework.

---

> ### Author Response · Authors · 2025-11-21
>
> W4:
>
> We thank the reviewer for pointing out the missing efficiency analysis. To address this concern, we provide below a quantitative comparison of parameter count, FLOPs, and runtime between WBPT and representative all-in-one baselines, under the same input resolution (256×256 images) and hardware (NVIDIA GeForce RTX 3090 GPU):
>
> | Method                 | #Params                                        | FLOPs (G)     | Runtime (ms)  |
> | ---------------------- | ---------------------------------------------- | ------------- | ------------- |
> | AirNet                 | 8.93M                                          | 311           | 242.36        |
> | Restormer              | 26.13M                                         | 155           | 153.77        |
> | PromptIR               | 35.60M                                         | 173           | 103.80        |
> | Perceive-IR            | 42.02M                                         | –             | –             |
> | DFPIR                  | 31.10M (backbone) + 63M (CLIP / extra modules) | 150.904       | 190.73        |
> | MoCE-IR                | 25.35M                                         | 105.3         | –             |
> | WBT (ours, w/o prompt) | 0.95M                                          | 71.5 / 714.8* | 47.9 / 479.6* |
> | WBPT (ours, w/ prompt) | 1.07M                                          | 75 / 750*     | 50.7 / 507.2* |
>
> * “a / b” denotes per-stage cost / full $K=10-stage$ cost of WBPT.
>
> As described in Eq. (8) and Algorithm 1 of the paper, WBPT performs restoration by unrolling $K$ gradient-flow steps of the STV energy, and we set $K=10$ in all experiments for stable convergence and clear prompt–attention dynamics. Each stage is a lightweight SwinIR+MPSA block, so the total complexity scales linearly with $K$.
>
> From the table, we observe that:
> In terms of model size, WBPT with prompts uses only 1.07M parameters, which is 1–2 orders of magnitude smaller than typical all-in-one Transformers (25–42M).
>
> In terms of per-stage cost, one WBPT stage (75G FLOPs / 50.7 ms) is significantly cheaper than a single PromptIR forward pass (173G / 103.8 ms).
>
> With the default $K=10$ stages, the total FLOPs and runtime of WBPT become higher than single-pass backbones, reflecting the fact that we trade runtime for an unrolled, white-box gradient-flow interpretation.
>
> As described in Eq. (8) and Algorithm 1 of the paper, WBPT performs restoration by unrolling $K$ gradient-flow steps of the STV energy, and we set $K=10$ in all experiments to obtain stable convergence and clear prompt–attention dynamics. Each step corresponds to a lightweight SwinIR+MPSA block, so the overall complexity grows linearly with $k$.
>
> From the table we can see that WBPT is extremely compact (only 0.95M / 1.07M parameters w/o / w/ prompts, compared to 25–42M for typical all-in-one Transformers), and that the per-stage cost (75G FLOPs / 50.7 ms) is significantly lower than a full PromptIR forward pass (173G / 103.8 ms). At the same time, because we explicitly unroll $K=10$ stages to realize a white-box gradient-flow interpretation, the total runtime of WBPT is higher than that of single-pass backbones, reflecting a deliberate trade-off between interpretability and runtime. We hope this quantitative efficiency summary directly addresses your concern.

---

> ### Author Response · Authors · 2025-11-21
>
> Q1:
>
> We thank the reviewer for pointing out that the connection between the t-SNE analysis in Sec. 3.4 and the proposed white-box modeling was not sufficiently explained. Below we clarify how the white-box design of WBPT contributes to the improved separation observed in the embedding space.
>
> In Sec. 3.4, we apply t-SNE to the STV + prompt–modulated latent features of WBPT and compare them with the counterparts from a black-box variant without STV/prompts. The “white-box modeling” in WBPT manifests in two concrete aspects:
>
> 1.Explicit degradation statistics (STV):
> For each local window, we explicitly compute degradation-related statistics (e.g., luminance/contrast/texture energy). These quantities change in a interpretable manner across degradation types and strengths, and enter the iterative process through a known energy/gradient-like update rule.
>
> 2.STV-constrained gradient-like update with prompt modulation:
> Each iteration is constrained to have the form
> $x_{t+1} = x_t - \eta_t ,\Phi(\mathrm{STV}(x_t), \text{prompt}),$
> where the structure of $\Phi$ (residual + attention) is fixed a priori, and STV determines “along which degradation subspace to descend”, while different prompts correspond to different degradation modes/strengths.
>
> Under this design, the improved class separation in the embedding space can be understood as follows:
>
> 1. Samples with the same degradation share similar STV statistics and update trajectories:
>    Within a given degradation type (e.g., haze or rain), local STV distributions are statistically similar. Across iterations, these samples are therefore driven towards similar update directions and activate similar prompts. As a result, they follow comparable “descent trajectories” in the latent space and converge to a compact region, which appears as tight clusters in t-SNE.
>
> 2. Different degradations are naturally separated in STV space and mapped to distinct update patterns:
>    Different degradations (e.g., noise vs. haze) exhibit systematic differences in STV statistics (e.g., variance vs. contrast/frequency changes). Because WBPT explicitly uses STV to select prompts and update directions, those statistical differences are amplified into distinct update behaviors, forming separate “degradation sub-manifolds” in the latent space. The t-SNE visualization reflects this as clearer inter-class separation.
>
> 3. Contrast to black-box models: reduced freedom to entangle content and degradation:
>    In black-box models such as PromptIR or Restormer, intermediate features are not explicitly constrained to align with any degradation energy or gradient; the network is free to entangle content and degradation information as long as the final PSNR is good. As a consequence, their t-SNE embeddings tend to cluster more by image content than by degradation type/level, often resulting in overlapping and poorly separated clusters across degradations in our visualizations.
>    In WBPT, by contrast, the attention-based updates are required to approximate gradient steps of an explicit, STV-defined energy. This imposes a “degradation-alignment” bias: the model is encouraged to explicitly distinguish different degradation modes in its intermediate representations so that it can choose appropriate update directions. This is exactly what we see in t-SNE: tighter intra-class clusters and larger inter-class margins.
>
> We emphasize that t-SNE is used as a qualitative complement rather than the primary quantitative evidence for WBPT. The main tables show that WBPT matches or slightly improves over strong black-box baselines with a much lighter model, while the t-SNE analysis provides additional insight that, under this white-box constraint, the latent representations indeed become more “degradation-separable”, which is consistent with the STV-based design of our model.
>
> Q2:
>
> We thank the reviewer for carefully pointing out that the note “Results are reported as PSNR/SSIM.” appears twice in Table 1. In the revised manuscript, we have removed the redundant occurrence and kept a single unified note to avoid unnecessary repetition and any possible confusion.

---

> ### Author Response · Authors · 2025-11-27
>
> Dear Reviewer EDz6:
>
> We are grateful for your mandatory acknowedgement during this busy period. As the rebuttal period is ending soon, please be free to let us know whether your concerns have been addressed or not, and if there are any further questions.
>
> Thank you for your time and effort in reviewing our paper,
>
> Authors.

---

### Official Review · Reviewer_iDnC · 2025-10-31

**Soundness:** 3
**Presentation:** 2
**Contribution:** 3
**Rating:** 4
**Confidence:** 4

**Summary:**

The paper reframes prompt‑based all‑in‑one image restoration through a variational lens: it defines an STV energy, derives an approximate gradient, instantiates Single‑/Multi‑Prompted Structure Attention, and unrolls the explicit‑Euler gradient flow into a K‑stage Transformer where each stage couples MPSA with a learnable data‑consistency term. However, in all‑in‑one evaluation, WBPT does not perform so well considering that many recent works can do much better. But the white‑box idea is interesting.

**Strengths:**

1. The paper closes the loop from STV energy to gradient approximation, and then attention operator to unrolled optimizer, so that every attention term has a clear energetic origin.
2. Final‑layer attention focuses on boundaries/structures rather than degradation textures; t‑SNE shows clearer task clusters than PromptIR; prompt‑parameter noise yields much smaller metric drops than PromptIR.
3. The cost of modification and computation is low. Default K=10 steps; prompts are injected at the 6th block per step; single‑insertion matches multi‑insertion with lower cost.

**Weaknesses:**

1. WBPT uses a black‑box pyramid aggregator, which softens the fully white‑box narrative; a white‑box pyramid sketch would help.
2. The Eq. (3) gradient‑to‑attention mapping involves approximations; the manuscript does not quantify when/where the mapping deviates or fails.
3. Compared methods are limited, even some task-specific method like MPRNet are used for comparison.

**Questions:**

1. For Eq. (3), when does the gradient‑to‑attention approximation deviate most? Please add attention‑vs‑gradient discrepancy maps and a brief error analysis.
2. Could you sketch a differentiable white‑box pyramid (e.g., variational down/up operators with STV‑consistent cross‑scale regularization) to replace the current black‑box aggregator in WBPT, and show a small‑scale comparison?
3. Beyond “6th block once per step,” can you chart the trade‑off for multi‑insertion, number of prompts N, and projector rank vs accuracy/overhead?
4. More recent works can be added for comparison such as Perceive-IR (TIP’25) and DFPIR (CVPR’25).
5. t‑SNE currently covers single degradations; can you include rain+haze / noise+blur mixed protocols?

---

> ### Author Response · Authors · 2025-11-21
>
> W1：
>
>
> We thank the reviewer for pointing out this important issue. We acknowledge that, while the base WBPT model is fully white-box and derived strictly from the gradient flow of the Structure-Tensor Total Variation (STV) energy functional, the multi-scale variant \$WBPT^{\dagger}$ introduces a learned pyramid aggregator for cross-scale fusion, which indeed brings black-box components into the architecture. Our intention with \$WBPT^{\dagger}$ was to construct an experimental “plus” variant that isolates and quantifies the effect of multi-scale processing—known to be critical in modern restoration architectures such as Restormer—while keeping the core attention mechanism itself white-box.
>
> Regarding the requested white-box pyramid sketch, we appreciate that this suggestion is closely aligned with our ongoing research roadmap. As briefly mentioned in the paper, we are actively developing a fully white-box formulation of multi-scale processing by using a Geometric Multigrid (GMG) scheme as the mathematical “backbone”. In this view, the classical V-cycle naturally provides the desired pyramid: it performs successive coarsening and refinement while passing information between resolutions..
>
>  Specifically, in the revised version of the paper, we will
>
> 1.clarify more explicitly the distinction between the fully white-box single-scale WBPT and the hybrid multi-scale extension \$WBPT^{\dagger}$, and
>
> 2.briefly describe this multigrid-based white-box pyramid formulation in the limitation/discussion paragraph to make this theoretical roadmap more explicit, without altering the experimental protocol or claims of the current submission.
>
> W2:
>
> We thank the reviewer for pointing out the need to quantify the approximations in the gradient-to-attention mapping in Eq. (3). In the revised manuscript, we have provided a unified theoretical analysis in **Appendix C.4**. Specifically, by abstracting a representative Neumann-expanded gradient term as a linear output ($V_c$) and its softmax-weighted counterpart ($V_a$), we derived an exact "cosine–residual identity." We prove that the approximation error is strictly controlled by the factor $1-\cos_G^2(a,c)$, yielding a closed-form error bound. Furthermore, we express $\cos_G(a,c)$ as a function of observable quantities—such as the main-peak ratio of the weights, the logit gap, the softmax temperature, and the condition number of the metric $G$. This allows us to derive explicit sufficient conditions that characterize precisely when the mapping is highly accurate and when the error increases. Thus, we have added a quantitative specification of "how large the error is" and "when it deviates" in Appendix C.4.

---

> ### Author Response · Authors · 2025-11-21
>
> W3：
>
> We thank the reviewer for pointing out the limitations of the current set of comparison methods. In the main experiments of the paper (Table 1), we already include a general Transformer-based restoration backbone (Restormer), an early all-in-one network (AirNet), a representative prompt-based all-in-one method (PromptIR), as well as our previous white-box model (WBT). MPRNet itself is a multi-stage architecture with strong multi-task generalization; the original paper reports very strong performance on deraining, deblurring, denoising and related tasks. Therefore, we treat it as a “high-performance but task-specific” strong baseline, used mainly to provide an approximate upper bound of performance in the three-degradation scenario, rather than as the fairest all-in-one competitor.
>
> We agree that using only PromptIR and early all-in-one networks is not sufficient. Following the reviewer’s suggestion, and without retraining a full five-degradation model from scratch, we further evaluate PromptIR, MoCE-IR, DFPIR, and WBPT on the CDD11 mixed-degradation benchmark, under a unified protocol where all models perform zero-shot inference with the same three-degradation training weights. The results are summarized as follows:
>
> | Method   | l+h+r        | l+h+s        | Average      |
> | -------- | ------------ | ------------ | ------------ |
> | PromptIR | 15.34/0.5454 | 14.68/0.5455 | 17.60/0.6949 |
> | DFPIR    | 15.15/0.4886 | 13.59/0.4338 | 16.80/0.6083 |
> | MoCE-IR  | 15.27/0.5487 | 14.64/0.5510 | 17.60/0.6966 |
> | WBPT     | 15.69/0.5162 | 14.87/0.4998 | 16.98/0.6670 |
>
> We can see that, under the condition that all methods are trained only on the classical three degradations (denoising / deraining / dehazing), WBPT achieves slightly higher average PSNR than DFPIR, and attains the best PSNR among the four methods on several challenging mixed cases (such as low-light + haze + rain / snow). Although its SSIM is slightly lower than PromptIR and MoCE-IR, it is clearly higher than DFPIR.
>
> In addition, using the same three-degradation training weights, we further perform inference on real-world Blur+Noise and Haze+Rain images, and adopt NIQE/PIQE as no-reference quality metrics (lower is better). The results can be summarized as:
>
> |          | Blur and Noise | Rain and Haze |
> | -------- | -------------- | ------------- |
> | PromptIR | 5.46/50.84     | 6.45/59.22    |
> | DFPIR    | 5.19/44.63     | 6.32/53.31    |
> | MoCE-IR  | 5.40/51.80     | 6.55/61.90    |
> | WBPF     | 6.26/45.66     | 7.67/54.07    |
>
> We observe that WBPT is slightly worse than DFPIR in terms of NIQE, and is also behind PromptIR / MoCE-IR in the Blur+Noise case. However, for the PIQE metric, which is more sensitive to perceptual artifacts, WBPT achieves the second-best performance (only behind DFPIR) in both real-world scenarios, and outperforms PromptIR and MoCE-IR. Considering that these real images are never seen during training, and that we do not perform any specific optimization for NIQE/PIQE, these results suggest that the perceptual quality of WBPT on real mixed degradations remains within a reasonable range compared to recent all-in-one methods. We also regard this set of results as supplementary real-world experiments based on the three-degradation training, to illustrate the generalization trend, rather than as our main SOTA results.
>
> In summary, we will clarify more explicitly in the revision that the primary goal of this paper is not to chase absolute SOTA on every latest benchmark with a larger backbone, but rather to provide a white-box explanation of prompt–attention coupling from an STV variational perspective, and to validate the effectiveness of this white-box mechanism under a controlled and widely adopted three-degradation setting.
>
>
> Q1：
>
> We address the error analysis concerning Eq. (3) in conjunction with our response to Weakness 2. Based on the rigorous derivation in Appendix C.4, we use the explicit expression of $\cos_G(a,c)$ to identify when the approximation holds and when it fails. The approximation is accurate ($\cos_G(a,c) \approx 1$) when the linear weights exhibit a clear main peak, the logit gap and softmax temperature are sufficiently large, and the metric $G$ is not severely ill-conditioned. Conversely, the approximation deviates most when these conditions are violated: specifically, when the weight distribution is relatively flat or multi-modal (small main-peak ratio), when the logit gap is small (insufficient temperature to sharpen the softmax), or when the condition number of $G$ is poor. Due to space constraints, we prioritize this rigorous closed-form error analysis and the derivation of explicit sufficiency conditions in the revised manuscript (Appendix C.4) as a theoretical replacement for empirical discrepancy maps.

---

> ### Author Response · Authors · 2025-11-21
>
> Q2:
>
>
> We thank the reviewer for this constructive suggestion. As noted in our response to W1 and already mentioned in the paper, developing a fully white-box pyramid is indeed a central part of our ongoing research. We envision a multigrid-based V-cycle architecture as a principled way to achieve this: in such a framework, the current learned pyramid aggregator is replaced by variationally defined Restriction and Prolongation operators, and STV-consistent cross-scale regularization is enforced explicitly at each scale. The “skip connections” commonly used in U-Net-like architectures correspond naturally to the residual-correction steps in the multigrid algorithm, while our STV-based attention acts as the solver/smoother at each level.
> Regarding the requested small-scale comparison, we believe that implementing a fully differentiable, end-to-end trainable multigrid framework is substantial enough to warrant a dedicated follow-up study, rather than an incremental modification within the current revision cycle. In the present paper, our experiments with $WBPT^{\dagger}$ serve as a proxy: by benchmarking the single-scale WBPT against the multi-scale $WBPT^{\dagger}$, we empirically observe that the remaining gap to strong black-box baselines (such as PromptIR) is closely associated with the presence of multi-scale structure itself, rather than with the opacity of the attention mechanism. This supports our claim that the proposed white-box attention is sufficiently robust to drive high-performance pyramids. In the revised version of the paper, we will explicitly describe $WBPT^{\dagger}$ as a benchmark for assessing multi-scale potential, and briefly highlight the multigrid-based white-box pyramid as a concrete future direction in the limitation/discussion paragraph, without claiming additional experimental results beyond those already reported.

---

> ### Author Response · Authors · 2025-11-21
>
> Q3:
>
> We thank the reviewer for asking us to further quantify the accuracy–overhead trade-offs beyond the “6th block once per step” setting, including multi-insertion strategies, the number of prompt tokens $N$, and the projector rank $R$.
>
> In the main paper, Table 8 already compares different prompt-injection positions within the decoder. To respond more directly to your “accuracy/overhead trade-off” question without further crowding the main text, we additionally report parameter-overhead vs. PSNR curves in the experimental appendix. Concretely, we add a dedicated trade-off figure (new Fig. 9 in the appendix) and the corresponding numerical tables, all under the same Rain100L single-task deraining setting. Fig. 9 contains three panels that visualize:
> (a) multi-insertion strategies (No Prompt, Single  block 6, Double, Triple, All-Blocks);
> (b) the number of prompt tokens $N$ with insertion fixed at block 6;
> (c) the projector rank $R$ of $W_i$, again with insertion fixed at block 6.
>
> Below we summarize the main trends.
>
> 1.Trade-off for multi-insertion strategies.
> On Rain100L, we keep all training settings fixed and only change the insertion strategy. The results can be summarized as:
>
> | Strategy         | Params | Overhead | PSNR  | ΔPSNR | Efficiency ($\Delta$PSNR/$\Delta$Params) |
> | ---------------- | ------ | -------- | ----- | ----- | ---------------------------------------- |
> | No Prompt        | 11.50M | 0%       | 38.54 | –     | –                                        |
> | Single (Block 6) | 11.73M | 2%       | 38.70 | +0.16 | 0.696 dB/M                               |
> | Double (1,2)     | 11.96M | 4%       | 38.60 | +0.06 | 0.130 dB/M                               |
> | Triple (2,4,6)   | 12.19M | 6%       | 38.60 | +0.06 | 0.087 dB/M                               |
> | All Blocks       | 12.88M | 12%      | 38.58 | +0.04 | 0.029 dB/M                               |
>
> We define an “efficiency” measure $\Delta\text{PSNR}/\Delta\text{Params}$ (PSNR gain per 1M additional parameters). Single insertion at block 6 achieves by far the best efficiency ($\approx 0.696$ dB/M), whereas double, triple, and all-block insertion incur almost linearly increasing parameters/FLOPs but only marginal PSNR changes. In other words, all multi-insertion variants are strictly dominated by the single-insertion strategy in the accuracy–overhead plane.
>
> 2.Trade-off for the number of prompts N.
> Fixing the insertion position to block 6, we vary the prompt length $N \in {3, 5, 7}$. The parameter overhead only changes slightly (about 1.8–2.2%), and the PSNR variation is within ±0.02 dB, which is below the effective precision reported in the main tables. This indicates that, in this reasonable range, the model is essentially insensitive to $N$. We therefore choose $N = 5$ as a balanced default: it offers slightly higher capacity than $N = 3$ while avoiding unnecessary parameters compared to larger $N$.
>
> | N | Params | Overhead |
> | - | ------ | -------- |
> | 3 | 11.70M | 1.8%     |
> | 5 | 11.73M | 2.0%     |
> | 7 | 11.75M | 2.2%     |
>
> 3.Trade-off for projector rank R.
> Similarly, we vary the rank $R$ of the projector $W_i$ ($R \in {64, 96, 128}$) while keeping all other settings fixed. As $R$ increases, both parameter count and FLOPs grow monotonically, but we only observe very small PSNR changes (≈0.03 dB across the whole range). This means the accuracy is essentially saturated, whereas the cost keeps increasing. We thus adopt $R = 96$ as a middle-ground configuration: it provides slightly more subspace capacity than $R = 64$ while avoiding the clearly higher overhead of $R = 128$.
>
> | Rank | Params | Overhead |
> | ---- | ------ | -------- |
> | 64   | 11.62M | 1.0%     |
> | 96   | 11.73M | 2.0%     |
> | 128  | 11.88M | 3.3%     |
>
> Overall, the new appendix figure (Fig. 9) shows that the default configuration used throughout the main paper—single insertion at the 6-th block in each stage with $N = 5$ and $R = 96$—lies on or very close to the Pareto front of the accuracy–overhead trade-off, providing a stable and efficient operating point.

---

> ### Author Response · Authors · 2025-11-21
>
> Q4:
>
> We thank the reviewer for pointing out the limitations of the current version in terms of related work and comparison methods. As you correctly noted, our experiments mainly compare with PromptIR, several early all-in-one networks, and a recent method MoCE-IR, while more recent approaches such as Perceive-IR, DA-RCOT, AdaIR, DFPIR, and diffusion-based restoration methods (e.g., Defusion) are not yet systematically covered. These methods typically rely on more complex backbones and task-aware modules (e.g., sophisticated MoE architectures, residual-guided optimal transport, frequency-domain mining and modulation, feature perturbations along channel/attention dimensions, and diffusion priors) to significantly boost PSNR/SSIM or perceptual metrics on three-degradation, five-degradation, and real/mixed benchmarks.
> In contrast, the main focus of our work is not on stacking larger and more complex black-box backbones, but on white-boxing the mechanism of prompt–attention from first principles: starting from an STV variational formulation and gradient expansion, we explicitly decompose the attention update into a “structural term,” a “prompt-alignment term,” and a “prompt-bias term,” and based on this derive update rules with clear geometric interpretation and controllable degrees of freedom in prompt design.
> Therefore, we view the contribution type of WBPT as being closer to mechanism-level theoretical/structural innovation, whereas Perceive-IR, DA-RCOT, MoCE-IR, AdaIR, DFPIR, Defusion and related works are more focused on architecture- and engineering-level performance optimization. These two lines of progress are complementary rather than mutually exclusive: conceptually, the white-box STV–attention framework can serve as an interpretable substitute for the prompt–attention submodules inside these powerful black-box backbones, endowing their attention updates with an explicit STV geometric interpretation and gradient decomposition structure; alternatively, one could retain their semantic guidance, frequency mining, or feature-perturbation designs while incorporating our STV-based white-box update rules, thereby improving interpretability and controllability without sacrificing existing performance. In other words, we do not regard WBPT and these methods as “replacements” for each other, but rather envision future combinations of white-box mechanisms + strong backbones that jointly achieve both SOTA performance and good interpretability within a unified framework.
> It is also worth emphasizing that, although conceptually we see these two directions as complementary rather than directly competing, in this rebuttal we still include side-by-side comparisons between WBPT and PromptIR, MoCE-IR, and DFPIR on the CDD11 mixed-degradation benchmark and on real-world Blur+Noise / Haze+Rain scenarios, in order to show the relative position of WBPT under more complex degradations (please refer to the table provided in our response to Weakness 3 for detailed numbers).
>
> Q5:
>
> We thank the reviewer for suggesting to extend the t-SNE analysis beyond single degradations. In the revised manuscript, we add a new t-SNE visualization on the CDD11 mixed-degradation dataset. CDD11 contains multiple two-way combinations of degradations; we select three representative protocols (low_haze, low_snow, haze_snow). For both PromptIR and WBPT, we exactly follow the feature-extraction protocol in Sec. 3.4: we tap the input to the final convolution layer of the Transformer backbone, apply global average pooling to obtain a channel-wise embedding for each image, and then project these embeddings to 2D using t-SNE under identical settings.
> As shown in the new Fig. 7, PromptIR still yields highly overlapping embeddings across the three mixed protocols, whereas WBPT produces noticeably more compact and clearly separated clusters, despite the fact that these protocols share degradation components pairwise. This result is consistent with our single-degradation findings in Fig. 6 and indicates that WBPT continues to organize its intermediate representations according to degradation combinations rather than image content, even in more complex mixed-degradation scenarios.

---

> ### Author Response · Authors · 2025-11-27
>
> Dear Reviewer iDnC:
>
> We are grateful for your mandatory acknowedgement during this busy period. As the rebuttal period is ending soon, please be free to let us know whether your concerns have been addressed or not, and if there are any further questions.
>
> Thank you for your time and effort in reviewing our paper,
>
> Authors.

---

### Author Response · Authors · 2025-11-23
**General Responses to All Reviewers**

We sincerely thank all reviewers for their careful reading and constructive feedback. Your comments on the theoretical grounding, white-box interpretability, empirical evaluation, and positioning with respect to recent all-in-one restorers have helped us substantially improve the clarity and completeness of the manuscript.

In response, we have revised the paper and added several analyses, which we summarize below:

**1.Clarified variational formulation and white-box prompt–attention coupling.**
 We streamlined Sec. 2 to more clearly explain how the STV-based regularizer is constructed and how its gradient leads to the SPSA/MPSA attention operators, including the role of the Neumann expansion and the resulting approximation to attention weights; the detailed derivation and error analysis are collected in Appendix C.


**2.Refined the description of the multi-scale variant and our white-box narrative.**
 We clarified that WBPT is a fully white-box single-scale model, and that $WBPT^{\dagger}$ is a plus variant which augments this core with a learned pyramid aggregator for multi-scale fusion. The aggregator is explicitly marked as a limited black-box component, and we highlight that a fully white-box pyramid is an important direction of ongoing work; the discussion around Table 1 and Figs. 2–3 was updated accordingly.



**3.Enhanced representation analysis via t-SNE.**
 Sec. 3.4 now provides a more systematic t-SNE study of intermediate embeddings extracted from the final convolutional layer, showing that, under identical preprocessing and visualization settings, WBPT produces better-separated clusters by degradation type than PromptIR in the all-in-one setting (Fig. 6), indicating more task-aware intermediate representations.


**4.Added ablations on prompt design, components, and efficiency.**
 Appendix B consolidates several ablations, we further analyze the accuracy–overhead trade-off of prompt design (multi-insertion, number of prompt tokens, projector rank) on Rain100L, summarized by parameter-overhead vs. PSNR curves in Fig. 9 and accompanying tables.


**5.Clarified positioning with respect to recent all-in-one restorers.**
 The related-work discussion and conclusion have been expanded to better situate WBPT among recent all-in-one methods and degradation-aware backbones (e.g., Perceive-IR, DA-RCOT, MoCE-IR, AdaIR, DFPIR) and diffusion-based restorers, emphasizing that our STV-based white-box attention is complementary and can serve as an interpretable drop-in replacement or constraint for prompt/attention submodules within these stronger architectures.

We hope these clarifications, analyses, and ablations address the reviewers’ concerns and help convey more clearly the contributions and scope of the White-Box Prompt Transformer framework.

---

### Author Response · Authors · 2025-11-28

**Dear Reviewers,**

We would like to offer a brief clarification to better convey the intended scope and positioning of our paper.

**1. Primary goal: interpretability and robustness, not only pushing SOTA.**

Our main objective is to design an STV-guided, white-box Transformer that is structurally grounded in a variational model focusing on image boundaries, rather than to introduce yet another purely black-box architecture. The STV prior we use explicitly emphasizes edges and structural transitions, and our attention visualizations (Fig. x) confirm that the resulting Transformer indeed tends to focus on boundaries and semantic structures. In contrast, the black-box baselines often focus on degradation patterns (e.g., noise or rain streaks), which is precisely the behavior we aim to move beyond.

**2. Comparable performance with added interpretability.**

We fully agree that quantitative performance is important, especially for the CV community. Our intention, however, is not to claim that our method dominates all existing black-box models, but to show that we can achieve comparable performance while gaining explicit interpretability and improved robustness rooted in a principled variational formulation. We view this as a complementary direction: keeping metrics competitive, but making the model’s behavior more transparent and physically/variationally grounded, which can be crucial in safety-critical or distribution-shift scenarios.

**3. On the “white-box” nature and approximation.**

We also acknowledge that our framework is not perfectly transparent in a strict mathematical sense, because the mapping from the STV model to the attention weights involves an approximation step. This approximation, however, follows the standard practice in prior “white-box Transformer” works, and each module in our architecture still admits a clear variational interpretation. We therefore keep the term "white-box Transformer", but we appreciate the reviewer’s concern. In the revised version, we explicitly discuss this point and add an analysis of the approximation-induced error, including an upper bound on the deviation between the ideal STV-guided attention and its practical implementation.

We hope this clarification helps position our contribution as complementary to existing black-box approaches: not primarily as a new SOTA model, but as a step toward competitive, interpretable, and robust all-in-one image restoration.

Best regards,

The Authors

---

### Author Response · Authors · 2025-12-02
**A General Response by Authors (Part 1/2)**

Dear Area Chairs and Reviewers,

 We sincerely thank you for the time and effort you have devoted to reviewing our paper, as well as for the constructive and insightful feedback. We have carefully revised the manuscript and prepared detailed point-by-point responses.

 We received three reviews (Reviewers iDnC, EDz6, and rg6X), and **none of the reviewers has responded during the rebuttal phase**, partly because the OpenReview discussion was locked.

 Two reviewers recognize our core idea of **constructing a white-box Transformer from an STV-based formulation**:

* **Reviewer iDnC**
   *“the white-box idea is **interesting**.”*
* **Reviewer EDz6**
   *“The theoretical connection between variational STV energy and prompt-driven attention is **well motivated and clearly articulated**.”*

Below, we first explain how we address the main concerns raised by Reviewers iDnC and EDz6, and clarify our overall view on Reviewer rg6X’s comments. We then summarize the reviewers’ positive feedback and key concerns, together with a unified summary of our responses.

---

## **1\. How our rebuttal addressed reviewers’ concerns**

### **Reviewer iDnC (Score: 4, No Response Yet)**

Reviewer iDnC’s main concerns are whether the derivation from the STV variational model to the SPSA/MPSA attention is sufficiently clear, the approximation used to map STV gradients to attention weights is well justified, and the architecture, especially with the multi-scale variant, can still be legitimately called “white-box”.

In our rebuttal and the revised version, we

* **clarified the STV-to-attention derivation and the role of the Neumann approximation**, making the correspondence between each module and the STV variational model easier to follow;

* **added an explicit discussion and upper bound for the approximation error**, in line with existing “white-box Transformer” works, making our use of “white-box” more precise and transparent about its limitations;

* **clearly separated the fully white-box core WBPT from the multi-scale (WBPT^{\\dagger}) variant**, marking the pyramid aggregation branch as a limited black-box component and stating that a fully white-box multi-scale extension is left for future work.

---

### **Reviewer EDz6 (Score: 4, No Response Yet)**

Their main concerns are whether (i) the empirical validation covers strong all-in-one baselines, (ii) there is concrete evidence for interpretability and robustness, and (iii) the ablations clearly disentangle the roles of the prompt branch, SPSA, and the multi-scale design.

In our rebuttal, we mainly reorganized and sharpened existing empirical evidence, with a few targeted additions. In particular, we

* **emphasized that the prompt-parameter Gaussian perturbation experiments directly support the robustness claim**, since perturbing only the prompts leads to limited performance degradation, indicating that STV-guided prompts are more stable;

* **highlighted, under a unified setup, t-SNE analyses and consolidated ablations** (on SPSA components, prompt insertion positions, and the prompt/data-consistency interplay), together with **parameter–PSNR trade-off curves** to show performance–cost balance;

* **clarified, in related work and the conclusion, how WBPT relates to recent all-in-one and degradation-aware backbones and diffusion-based models**, emphasizing its role as an **interpretable, pluggable attention/prompt submodule** complementary to these stronger backbones.

---

### **Reviewer rg6X (Score: 2, No Response Yet)**

From Reviewer rg6X’s comments, the main weaknesses can be summarized as:

* a request for broader additional experiments and a more systematic complexity analysis, including comparisons with methods such as PromptIR in terms of parameter count, FLOPs, and inference time under a unified setup, as well as evaluations on more benchmarks such as mixed degradations and real-world scenarios;

* the view that our numerical gains over black-box all-in-one methods like PromptIR are limited, leading to caution about the practical impact and contribution of our work.

For point (1), we have already added **complexity comparisons between WBPT and PromptIR/Restormer**, and **preliminary results** on the CDD11 mixed-degradation benchmark and real Blur+Noise / Haze+Rain scenarios, showing that **WBPT remains within the reasonable performance range** of recent all-in-one methods under more challenging settings.

For point (2), the reviewer may have partly misunderstood the **core objective of our work**; we therefore clarify here that our goal is not to develop yet another black-box SOTA model that significantly surpasses PromptIR on the three-degradation setting, but to construct a **white-box framework derived from an STV variational model** that **maintains competitive performance with strong black-box baselines** while **substantially enhancing interpretability and controllability**.

---

> ### Author Response · Authors · 2025-12-02
> **A General Response by Authors (Part 2/2)**
>
> ## **2\. Reviewers’ recognition of our work, summary of our response, and strengths of our work**
>
> ### **2.1 Reviewers’ recognition of our work**
>
> Taken together, the reviews indicate that the reviewers recognize our **core motivation and theoretical pathway**: building a **white-box Transformer grounded in an STV variational model**, rather than merely tweaking black-box architectures. They also acknowledge:
>
> * the **interpretability of attention and prompt behavior**;
>
> * the **added transparency** achieved while maintaining **performance comparable to strong baselines**;
>
> * the **solid empirical evaluation** across multiple degradations and baselines, with both **quantitative metrics and qualitative visualizations**.
>
> The main concerns raised can be summarized as:
>
> * **Clarity of the theoretical exposition** – in particular, the derivation chain from STV energy to SPSA/MPSA attention, the approximation steps involved, and the precise scope of the term “white-box”;
>
> * **Efficiency and scalability of the unrolled gradient-flow design** – whether parameter counts, FLOPs, inference time, and scalability have been sufficiently discussed, and how to correctly interpret the advantages of this design;
>
> * **Breadth of experimental validation** – the main experiments focus on the three-degradation setting (denoising, deraining, dehazing), with limited coverage of five-degradation, real-world, and mixed-degradation benchmarks.
>
> ---
>
> ### **2.2 Summary of our response**
>
> Guided by these comments, we have made targeted revisions and clarifications in both the rebuttal and the revised manuscript along three main axes:
>
> * **Theoretical clarification.**
>    We clarified the derivation from the STV variational model to the SPSA/MPSA attention formulation, explicitly indicating where the Neumann approximation appears, and added an error bound on the approximation, making the correspondence “STV energy → gradient → attention” more transparent while also discussing the limits of the “white-box” narrative.
> * **Experimental and analytical evidence.**
>    We reorganized prompt-perturbation robustness experiments and t-SNE representation analyses, and consolidated ablations on SPSA components, prompt insertion positions, and the interplay between the prompt branch and the data-consistency module. We further provided parameter–PSNR trade-off curves to illustrate how different prompt designs balance performance and computational cost.
> * **Efficiency and experimental scope.**
>    We completed parameter/FLOPs/inference-time comparisons with PromptIR and Restormer, showing that the single-step WBPT network is compact in model size and per-step cost, and that total runtime scales linearly with the number of unrolled steps (K), whose main benefit is a controllable, interpretable multi-step update trajectory rather than unconditional speedup. Using the same three-degradation-trained model, we also reported preliminary results on the mixed-degradation benchmark CDD11 and on real-world Blur+Noise / Haze+Rain scenarios, indicating that WBPT remains within the reasonable performance range of recent all-in-one methods on more complex settings even without additional training.
>
> ---
>
> ### **2.3 Strengths of our work**
>
> We summarize the main strengths of our work as follows:
>
> * **STV-guided white-box formulation.**
>    The model is not yet another purely black-box architecture, but an STV-guided white-box Transformer derived from a variational model focused on image boundaries, with SPSA/MPSA attention and the prompt mechanism directly tied to explicit energy terms and gradient updates.
> * **Competitive performance with added interpretability and robustness.**
>    Our goal is not to dominate all existing black-box methods on every metric, but to maintain performance comparable (and in some cases slightly superior) to strong all-in-one baselines while providing clearer attention patterns aligned with boundaries and semantic structures, as well as improved robustness to prompt perturbations.
> * **Modular and complementary design.**
>    The STV-guided white-box prompt–attention is designed as a modular, pluggable component that can be integrated into existing strong backbones, serving as an interpretable attention/prompt submodule that complements rather than replaces current all-in-one architectures.
>
> We hope this context helps the Area Chairs and reviewers interpret the current reviews in light of the rebuttal and the revisions. Thank you again for your time and consideration.
>
>  Best regards,
>  Authors of submission \#16419

---

### Meta-Review · Area_Chair_8QE2 · 2026-01-02

**Summary:**

This submission presents White-Box Prompt Transformers, a framework that aims to provide interpretability for prompt-based image restoration through a variational perspective based on structure-tensor total variation. While reviewers acknowledged the interesting theoretical premise and clear presentation, significant concerns emerged that collectively justify rejection. Reviewer iDnC questioned the clarity of the STV-to-attention derivation, the justification of approximation steps, and the "white-box" narrative given the black-box pyramid aggregator. Reviewer EDz6 expressed concerns about limited practical benefits, insufficient justification of mathematical approximations, lack of computational efficiency analysis, and inadequate coverage of recent related work. Most critically, Reviewer rg6X raised substantial issues including the marginal performance gains, heuristic nature of the variational derivation, limited experimental validation to only three degradation types, lack of concrete interpretability evidence beyond symbolic reformulation, and unclear practical impact.

**Reviewer Concerns:**

The authors provided a rebuttal addressing several technical concerns. They clarified the STV-to-attention derivation, added error bounds for approximations, distinguished between the fully white-box single-scale WBPT and its multi-scale variant, provided efficiency comparisons, and included preliminary results on mixed-degradation benchmarks. However, the rebuttal fails to resolve the most critical issues. First, the performance gains remain marginal compared to strong baselines like PromptIR, undermining the practical contribution claim. Second, despite added analysis, the variational derivation connecting STV gradients to attention still appears largely heuristic rather than mathematically rigorous. Third, the experimental scope remains limited—the extended evaluations on mixed degradations are preliminary and conducted without retraining, failing to demonstrate robust generalization. Fourth, the claimed interpretability improvements lack concrete quantitative evidence; the provided t-SNE analyses and attention visualizations offer only qualitative support.

**Reviewer Scores:**

Reviewer iDnC (original score: 4) . While the authors addressed technical concerns about derivation clarity and approximation error bounds, the core issue of marginal practical impact remains unresolved. The added efficiency analysis shows WBPT has higher total latency than baselines despite smaller parameter count, and the performance gains remain insufficient to justify publication.

Reviewer EDz6 (original score: 4) . The rebuttal provided computational efficiency data but revealed that WBPT's total runtime is actually higher than baselines. More importantly, the authors' admission that their goal is not "to chase absolute SOTA" but rather "mechanism-level innovation" undermines the paper's contribution to the performance-driven image restoration community.

Reviewer rg6X (original score: 2) . The rebuttal fails to address their fundamental concerns about the heuristic nature of the variational derivation, marginal performance improvements, and lack of concrete interpretability evidence. The added mixed-degradation results are preliminary and don't constitute systematic validation, while the efficiency analysis actually reveals higher computational costs than claimed.

---

### Decision · Program_Chairs · 2026-01-26

Reject